# Independent theta phase coding accounts for CA1 population sequences and enables flexible remapping

**Angus Chadwick[1,2], Mark CW van Rossum[1], Matthew F Nolan[3]***

[1]Institute for Adaptive and Neural Computation, School of Informatics, University of Edinburgh, Edinburgh, United Kingdom; [2]Neuroinformatics Doctoral Training Centre, School of Informatics, University of Edinburgh, Edinburgh, United Kingdom; [3]Centre for Integrative Physiology, University of Edinburgh, Edinburgh, United Kingdom

**Abstract** Hippocampal place cells encode an animal's past, current, and future location through sequences of action potentials generated within each cycle of the network theta rhythm. These sequential representations have been suggested to result from temporally coordinated synaptic interactions within and between cell assemblies. Instead, we find through simulations and analysis of experimental data that rate and phase coding in independent neurons is sufficient to explain the organization of CA1 population activity during theta states. We show that CA1 population activity can be described as an evolving traveling wave that exhibits phase coding, rate coding, spike sequences and that generates an emergent population theta rhythm. We identify measures of global remapping and intracellular theta dynamics as critical for distinguishing mechanisms for pacemaking and coordination of sequential population activity. Our analysis suggests that, unlike synaptically coupled assemblies, independent neurons flexibly generate sequential population activity within the duration of a single theta cycle.

**\*For correspondence:** mattnolan@ed.ac.uk

**Competing interests:** The authors declare that no competing interests exist.

**Reviewing editor**: Frances K Skinner, University Health Network, and University of Toronto, Canada

## Introduction

Cognitive processes are thought to involve the organization of neuronal activity into phase sequences, reflecting sequential activation of different cell assemblies (*Hebb, 1949*; *Harris, 2005*; *Buzsáki, 2010*; *Wallace and Kerr, 2010*; *Palm et al., 2014*). During navigation, populations of place cells in the CA1 region of the hippocampus generate phase sequences structured around the theta rhythm (e.g., *Skaggs et al., 1996*; *Dragoi and Buzsáki, 2006*; *Foster and Wilson, 2007*). As an animal moves through the firing field of a single CA1 neuron, there is an advance in the phase of its action potentials relative to the extracellular theta cycle (*O'Keefe and Recce, 1993*). Thus, populations of CA1 neurons active at a particular phase of theta encode the animal's recent, current, or future positions (*Figure 1A,B*). One explanation for these observations is that synaptic output from an active cell assembly ensures its other members are synchronously activated and in addition drives subsequent activation of different assemblies to generate a phase sequence (*Figure 1C*) (*Harris, 2005*). We refer to this as the *coordinated assembly hypothesis*. An alternative possibility is that independent single cell coding is sufficient to account for population activity. According to this hypothesis, currently active assemblies do not determine the identity of future assemblies (*Figure 1D*). We refer to this as the *independent coding hypothesis*.

Since these coding schemes lead to different views on the nature of the information transferred from hippocampus to neocortex and on the role of CA1 during theta states, it is important to distinguish between them. While considerable experimental evidence has been suggested to support the coordinated assembly hypothesis (e.g., *Harris et al., 2003*; *Dragoi and Buzsáki, 2006*; *Foster*

**eLife digest** When we explore a new place, we naturally create a mental map of the location as we go. This mental map is stored in a region of the brain called the hippocampus, which contains cells called place cells. These cells can carry information about our past, present, and future location in the form of electrical signals. They connect to each other to form networks and it has been proposed that these connections can store the information needed for the mental maps.

Real-time maps are represented in the information carried by the electrical signals themselves. A physical location is specified by the individual place cell that is activated, and by the timing of the electrical signal it produces relative to a 'brain wave' called the theta rhythm. Brain waves are patterns of electrical signals activated in sets of brain cells and the theta rhythm is produced in the hippocampus of an animal as it explores its surroundings.

Previous experiments suggested that when a rat explores an area, several sets of brain cells in the hippocampus are activated in sequence within each cycle of the theta rhythm. As the rat moves forward, the sequence shifts to different sets of cells to reflect the upcoming locations ahead of the rat. It has been thought that these sequences are triggered by the individual connections between the place cells.

Here, Chadwick et al. developed mathematical models of the electrical activity in the brains of rats as they explored. They used these models to analyze data from previous experiments and found that the sequences of electrical activity arise from the timing of each cell's activity relative to the theta rhythm, rather than from the connections between the cells.

Chadwick et al.'s findings suggest that the mental map may be highly flexible, allowing vast numbers of distinct memories to be stored within the same network of place cells without interference. Future studies will involve investigating the role of brain waves in the forming new mental maps and creating new memories.

*and Wilson, 2007*; *Maurer et al., 2012*; *Gupta et al., 2012*), the extent to which complex sequences of activity in large neuronal populations can be accounted for by independent coding is not clear. To address this we developed phenomenological models of independent and coordinated place cell activity during navigation. In the independent coding model, the spiking activity of each cell is generated by rate coding across its place field and phase precession against a fixed theta rhythm. We show that in this model phase coding generates a traveling wave which propagates through the population to form spike sequences. This wave is constrained by a slower moving modulatory envelope which generates spatially localized place fields. In the coordinated assembly model, the spikes generated by each cell are also influenced by the activity of other cells in the population. As a result, population spike patterns are further entrained by population interactions which counter the effects of single cell spike time variability and increase the robustness of theta sequences.

The independent coding hypothesis predicts that a population of independent cells will be sufficient to explain the spatiotemporal dynamics of cell assemblies in CA1. In contrast, the coordinated assembly hypothesis predicts that groups of cells show additional coordination beyond that imposed by a fixed firing rate and phase code (*Harris et al., 2003*; *Harris, 2005*). We show that the independent coding model is sufficient to replicate experimental data previously interpreted as evidence for the coordinated assembly hypothesis (*Harris et al., 2003*; *Dragoi and Buzsáki, 2006*; *Foster and Wilson, 2007*; *Maurer et al., 2012*; *Gupta et al., 2012*), despite the absence of coordination within or between assemblies. Moreover, novel analyses of experimental data support the hypothesis that place cells in CA1 code independently. Independent coding leads to new and experimentally testable predictions for membrane potential oscillations and place field remapping that distinguish circuit mechanisms underlying theta sequences. In addition we show that, despite the apparent advantage of coordinated coding in generating robust sequential activity patterns, it suffers from an inability to maintain these patterns in a novel environment. Thus, a key advantage of sequence generation through independent coding is to allow flexible global remapping of population activity while maintaining the ability to generate coherent theta sequences in multiple environments.

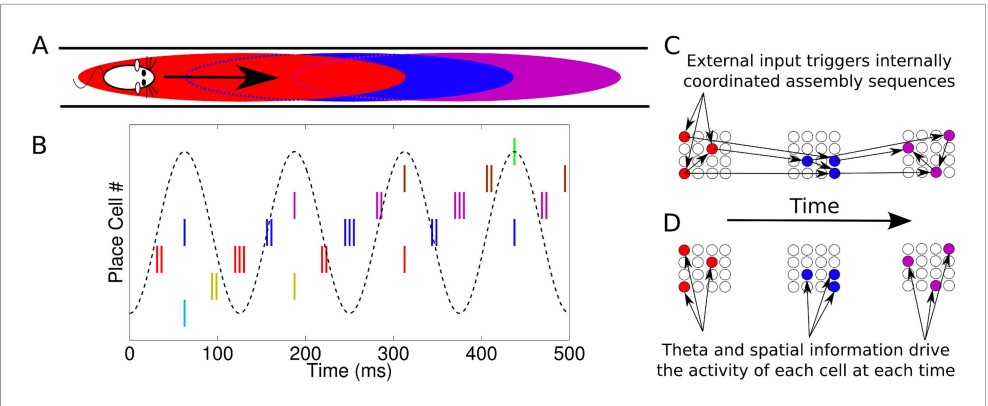

**Figure 1**. Phase sequences in a place cell population. (**A**) During navigation, place cells are sequentially activated along a route. (**B**) Within each theta cycle, this slow behavioral sequence of place cell activations is played out on a compressed timescale as a theta sequence. Theta sequences involve both rate and phase modulation of individual cells, but it remains unclear whether additional coordination between cells is present. (**C**) Internal coordination may bind CA1 cells into assemblies, and sequential assemblies may be chained together synaptically. This would require specific inter- and intra-assembly patterns of synaptic connectivity within the network. (**D**) Alternatively, according to the independent coding hypothesis, each cell is governed by theta phase precession without additional coordination. DOI: 10.7554/eLife.03542.003

## Results

### Single cell coding model

To test the independent coding hypothesis, we developed a phenomenological model which generates activity patterns for place cell populations during navigation. While a phenomenological model of CA1 phase precession has previously been developed (*Geisler et al., 2010*), several features of this model limit its utility for investigation of coordination across neuronal populations. First, the previous model addresses only the temporal dynamics of single unit activity and population average activity, without addressing the spatiotemporal patterns of spiking activity within the population, the nature of which is a central question in the present study. Second, the previous model assumes coordination between cells in the form of fixed temporal delays and is formulated for a fixed running speed. In contrast, we wish to understand in detail the temporal relationships between cells arising in populations with no direct coordination and how these temporal relationships might depend on factors such as running speed. We therefore develop a model of a single cell with a given place field and phase code and proceed to derive the patterns of population activity under the independent coding hypothesis. To do this, we modeled the firing rate field for each neuron using a Gaussian tuning curve:

$$r_x(x) = A \exp\left(-\frac{(x - x_c)^2}{2\sigma^2}\right), \tag{1}$$

where $r_x$ describes firing rate when the animal is at location $x$ within a place field with center $x_c$, width $\sigma$, and maximum rate $A$ (*Figure 2A*, top panel). Simultaneously, we modeled the firing phase using a circular Gaussian:

$$r_\phi(\phi(x), \theta(t)) = \exp\left(k \cos(\phi(x) - \theta(t))\right), \tag{2}$$

where $r_\phi$ describes the firing probability of the neuron at each theta phase at a given location (*Figure 2B*). Here, $\theta(t) = 2\pi f_\theta t$ is the local field potential (LFP) theta phase at time $t$ and $\phi(x)$ is the preferred firing phase associated with the animal's location $x$, termed the *encoded phase*. The encoded phase $\phi(x)$ is defined to process linearly across the place field (*Figure 2A*, bottom panel; *Supplementary file 1*, Appendix: A1). The phase locking parameter $k$ determines the precision at which the encoded phase is represented in the spike output (*Figure 2B*). The instantaneous firing rate of the cell is given by the product of these two components $r = r_x r_\phi$. The phase locking can be set so that the cell exhibits only rate coding (at $k = 0$, where $r = r_x$), only phase coding (as $k \to \infty$, where all spikes occur at exactly the encoded phase $\phi(x)$) or anywhere in between (*Figure 2C*).

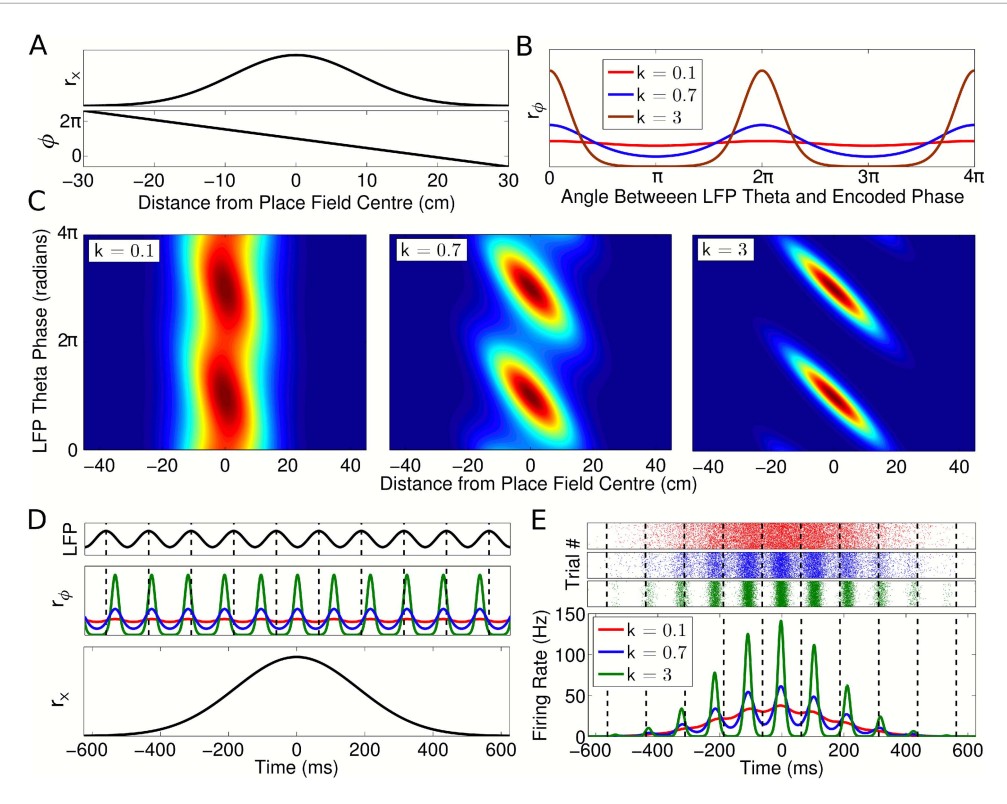

**Figure 2.** Single cell coding model. (**A**) Firing rate and phase at different locations within a cell's place field are determined by a Gaussian tuning curve $r_x$ and linearly precessing encoded phase $\phi$, respectively. (**B**) The dependence of single cell activity on the LFP theta phase $\theta$ is modeled by a second tuning curve $r_\phi$ which depends on the angle between the LFP theta phase $\theta$ and encoded phase $\phi$ at the animal's location. The phase locking parameter $k$ controls the precision of the phase code. (**C**) The combined dependence of single cell activity on location and LFP theta phase. (**D**) Temporal evolution of the rate and phase tuning curves for a single cell as a rat passes through the place field at constant speed. (**E**) The total firing rate corresponding to (**D**), and spiking activity on 1000 identical runs.

The following figure supplement is available for figure 2:

**Figure supplement 1**. Effect of normalization factor ($N_{spikes}$).

To model place cell activity during navigation on a linear track, we set $x(t) = vt$, where $v$ is the running speed (*Figure 2D,E*). This causes the encoded phase $\phi(t)$ to precess linearly in time at a rate $f_\phi$ which is directly proportional to running speed and inversely proportional to place field size, as in experimental data (*Huxter et al., 2003*; *Geisler et al., 2007*). To generate spikes we used an inhomogeneous Poisson process with an instantaneous rate $r = r_x r_\phi$. We normalized the firing rate such that the average number of spikes fired on a pass through a place field is independent of running speed (see *Supplementary file 1*, Appendix: A2) (*Huxter et al., 2003*). If the phase $\phi(x)$ at each location in the place field is fixed, the full rate and phase coding properties of a cell are encompassed by three independent parameters—the width of the spatial tuning curve $\sigma$, the degree of phase locking $k$, and the average number of spikes per pass $N_{spikes}$. Phase precession (*Figure 2C*) and firing rate modulation as a function of time in this model (*Figure 2E*) closely resemble experimental observations (e.g., *Skaggs et al., 1996*; *Mizuseki and Buzsaki, 2013*).

Place cells often show variations in firing rate in response to nonspatial factors relevant to a particular task (e.g., *Wood et al., 2000*; *Fyhn et al., 2007*; *Griffin et al., 2007*; *Allen et al., 2012*). In our model, such multiplexing of additional rate coded information can be achieved by varying the number of spikes per pass $N_{spikes}$ without interfering with the other parameters $\phi(x)$, $\sigma$, and $k$ (*Figure 2—figure supplement 1*).

It has been shown that the trial to trial properties of phase precession in individual cells are more variable than would be expected based on the pooled phase precession data (*Schmidt et al., 2009*). While it is possible that such trial to trial variability could reflect coordination between cell assemblies, such variability is equally consistent with an independent population code, and our model can be readily extended to incorporate such properties (*Supplementary file 1*, Appendix: A2).

## Independent phase coding generates traveling waves

Given this single cell model and assuming an independent population code, we next investigated the spatially distributed patterns of spiking activity generated in a CA1 population. To map the spatiotemporal dynamics of the population activity onto the physical space navigated by the animal, we analyzed the distributions of the rate components $r_x$ and phase components $r_\phi$ of activity in cell populations sorted according to the location $x_c$ of each place field (*Supplementary file 1*, Appendix: A3).

Our model naturally generates population activity at two different timescales: the slow behavioral timescale at which the rat navigates through space and a fast theta timescale at which trajectories are compressed into theta sequences. While the rat moves through the environment, the spatial tuning curves $r_x(x)$ generate a slow moving 'bump' of activity which, by definition, is comoving with the rat (*Figure 3A*, top, black). Simultaneously, the phasic component $r_\phi(\phi(x),\theta(t))$ instantiates a traveling wave (*Figure 3A*, top, red). Due to the precession of $\phi(t)$, the wave propagates forward through the network at a speed faster than the bump, resulting in sequential activation of cells along a trajectory on a compressed timescale. The slower bump of activity acts as an envelope for the traveling wave, limiting its spatial extent to one place field (*Figure 3A*, bottom). The continuous forward movement of

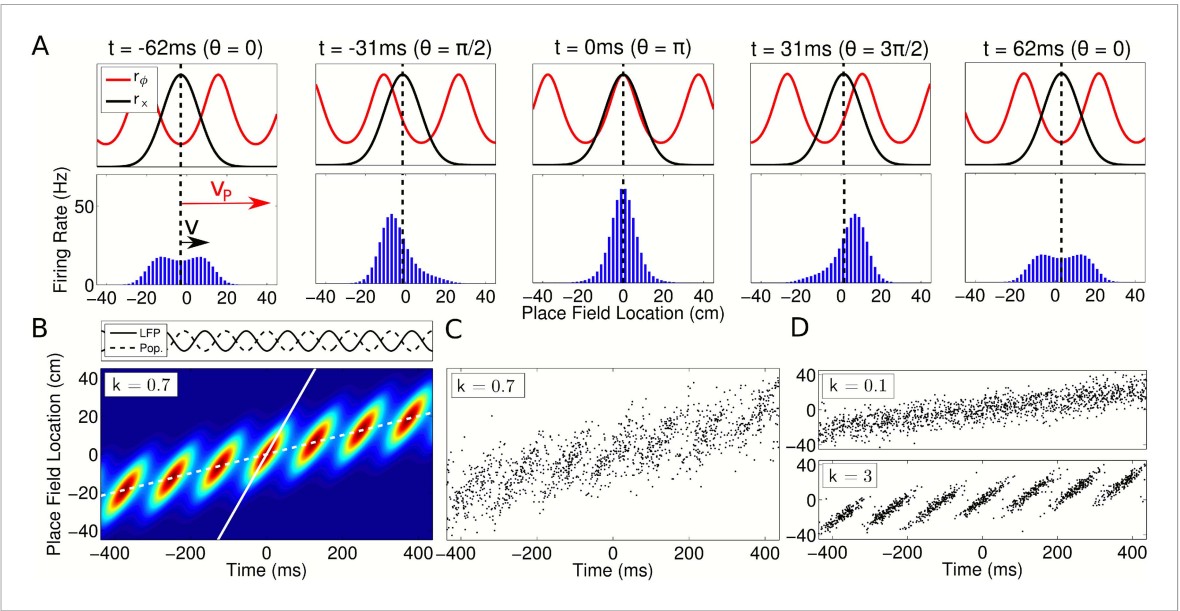

**Figure 3**. Spatiotemporal dynamics of CA1 populations governed by independent coding. (**A**) Top: Population dynamics during a single theta cycle on a linear track after ordering cells according to their place field center $x_c$ in physical space. The two components of the population activity are shown—the slow moving envelope (black) and the fast moving traveling wave (red), which give rise to rate coding and phase coding, respectively (cf. *Figure 2*). Bottom: Resulting firing rates across the population. When the traveling wave and envelope are aligned, the population activity is highest (middle panel). The dashed line shows the location of the rat at each instant. (**B**) Firing rate in the population over seven consecutive theta cycles. The fast and slow slopes are shown (solid and dashed lines, respectively), corresponding to the speeds of the traveling wave and envelope as shown in part (**A**). The top panel shows the LFP theta oscillations and emergent population theta oscillations, which are generated by the changing population activity as the traveling wave shifts in phase relative to the slower envelope (see *Video 1*). (**C** and **D**) The spiking activity for a population of 180 cells. All panels used $v = 50$ cm/s, so that $v_p = 350$ cm/s and $c = 7$.
The following figure supplement is available for figure 3:

**Figure supplement 1**. CA1 population activity governed by coordinated assemblies.

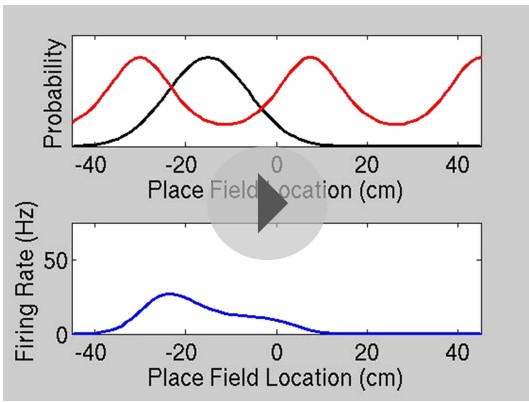

**Video 1.** Traveling wave dynamics in populations of CA1 place cells. Top: Distribution of the rate (black) and phasic (red) tuning curves for a population of linear phase coding place cells during constant speed locomotion on a linear track (cf. *Figure 3A*). The evolution in the population over 7 consecutive theta cycles is shown, slowed by a factor of approximately 16x. Bottom: The evolution of the overall firing rate distribution in the population, generated by multiplying the two tuning curves shown in the top panel. Note that the population firing rate undergoes oscillations at LFP theta frequency and the center of mass of the population activity shifts from behind the animal to ahead of the animal in each theta cycle.

the traveling wave is translated into discrete, repeating theta sequences via a shifting phase relationship to the slow moving component (*Figure 3B–D*, *Video 1*). Moreover, this shifting phase relationship generates global theta oscillations at exactly the LFP frequency that cells were defined to precess against (*Figure 3B*, top panel). Thus, our model can be recast in terms of the dynamics of a propagating wave-packet comprising two components, with network theta resulting from their interaction. While we define single cells to precess against a reference theta rhythm (i.e., the LFP), we now see that this same reference oscillation emerges from the population, despite higher frequencies of individual cells.

Our model's prediction of global theta oscillations emerging in networks of faster oscillating place cells is consistent with a previous phenomenological model which assumed a fixed running speed and fixed, experimentally determined temporal delays between cells (*Geisler et al., 2010*). However, in contrast to previous models, our model based on single cell coding principles allows an analysis in which only place field configurations and navigational trajectories are required to fully predict at any running speed both the global theta oscillation and the detailed population dynamics. Experimental data show that the frequency of LFP theta oscillations is relatively insensitive to the running speed of the animal, showing a mild increase with running speed compared to a larger single unit increase (*Geisler et al., 2007*). We therefore investigated the relationship between the running speed of the animal, the temporal delays between cells and the frequency of population theta oscillations in the independent coding model.

The spiking delays between cells in our model are determined by speed of the fast moving traveling wave $v_p$, which is related to the rat's running speed $v$ by:

$$v_p = cv, \tag{3}$$

where $c$ is called the *compression factor*. This factor is equivalent to the ratio of the rat's actual velocity and the velocity of the representation within a theta cycle and has been quantified in previous experimental work (*Skaggs et al., 1996*; *Dragoi and Buzsáki, 2006*; *Geisler et al., 2007*; *Maurer et al., 2012*), although the relationship to the traveling wave model developed here was not previously identified (see *Supplementary file 1*, Appendix: A2 for derivation).

Analysis of our model demonstrates that for an independent population code the compression factor naturally depends on running speed. This change in compression factor with running speed ensures that the network maintains a fixed population theta frequency while running speed and single unit frequency vary:

$$v_p - v = \lambda f_\theta, \tag{4}$$

where the constant $\lambda$ is the wavelength of the traveling wave (equal to the size of a place field, measured as the distance over which a full cycle of phase is precessed [*Maurer et al., 2006*]) and $v_p - v$ stays constant across running speeds due to the changing compression factor.

Hence, independent coding predicts temporal delays which are dependent on running speed. Conversely, our analysis shows that models incorporating fixed temporal delays between cells (e.g., *Diba and Buzsáki, 2008*; *Geisler et al., 2010*) cannot maintain an invariant relationship between

spike phase and location without producing a population theta oscillation whose frequency decreases rapidly with running speed, in conflict with experimental observations (*Geisler et al., 2007*).

## Assembly coordination stabilizes sequential activation patterns

In order to compare activity patterns predicted by independent coding schemes with those predicted when interactions between cell assemblies are present, we developed a second model in which the spiking activity of each place cell influences the spiking activity of peer cells within the population. While single cell rate and phase tuning curves in this coordinated assembly model are identical to those in the independent coding model, a peer weight function also modulates the probability of a spike occurring in each cell depending on the spikes of its peers (*Figure 3—figure supplement 1A*, *Supplementary file 1*, Appendix: A4). In this model, asymmetric excitation stabilizes the temporal relationship between sequentially activated assemblies, while feedback inhibition between place cells normalizes firing rates (cf. *Tsodyks et al., 1996*). The resulting sequences are considerably more robust than those generated by independent coding with the same single cell properties (*Figure 3—figure supplement 1B–C*). Assembly interactions also amplify theta oscillations in the network (*Figure 3—figure supplement 1D*) (*Stark et al., 2013*). Hence, assembly coordination provides a potential mechanism for stabilizing the sequential activity patterns generated by noisy neurons, as interactions entrain cells in the population into coherent activation patterns within each theta cycle.

While alternative forms of assembly coordination might also be considered, we choose the present model for two key reasons. First, this model is simple, containing relatively few adjustable parameters while capturing the essential features of sequence generation via assembly coordination. Second, as we will show below, the coordination between cells under this model is sufficient to evaluate statistical tests of independence, allowing a systematic framework with which to interpret the results of such tests on experimental data.

## Independent coding accounts for apparent peer-dependence of CA1 activity

We next investigated the extent to which models for population activity based on independent coding and coordinated assemblies can account for observations previously suggested to imply coordination within and between assemblies (*Harris et al., 2003*; *Dragoi and Buzsáki, 2006*; *Foster and Wilson, 2007*; *Maurer et al., 2012*; *Gupta et al., 2012*). We show below that, although these observations at first appear to imply assembly coordination, they can be accounted for by the independent coding model. We go on to establish the power of several tests to distinguish spike patterns generated by independent and coordinated coding models. By applying these tests to experimental data, we provide further evidence that CA1 population activity is generated through independent coding.

We first assessed whether independent coding accounts for membership of cell assemblies. A useful measure of the coding properties of place cell populations is to test how accurately single unit activity can be predicted from different variables. If, after accounting for all known single cell coding properties, predictions of the activity of individual place cells can be further improved by information about firing by their peer cells, it is likely that such cells are interacting through cell assemblies (*Harris, 2005*). Initial analysis of CA1 place cell firing suggested this is the case, with coordination between cells at the gamma timescale being implicated (*Harris et al., 2003*). Because this improved predictability directly implies interactions between CA1 neurons, it would constitute strong evidence against the independent coding hypothesis. However, in accounting for single cell phase coding properties, the prediction analysis of *Harris et al. (2003)* assumed that firing phase is independent of movement direction in an open environment. In contrast, more recent experimental data show that in open environments firing phase always precesses from late to early phases of theta, so that firing phase at a specific location depends on the direction of travel (*Huxter et al., 2008*; *Climer et al., 2013*; *Jeewajee et al., 2014*). Therefore, to test if the apparent peer-dependence of place cell activity is in fact consistent with independent coding, the directionality of phase fields must be accounted for.

To address this we first considered whether the assumption of a nondirectional phase field would lead to an erroneous conclusion of coordinated coding when analyzing spike patterns generated by the independent coding model. To do this, we extended the traveling wave model to account for phase precession in open environments (*Supplementary file 1*, Appendix: A6). We then constructed

phase fields from simulated spiking data following the approach of *Harris et al. (2003)*, in which firing phase is averaged over all running directions, and separately constructed directional phase fields consistent with recent experimental observations (*Huxter et al., 2008*; *Climer et al., 2013*; *Jeewajee et al., 2014*). We then calculated the predictability of neuronal firing patterns generated by the independent coding model using each of these phase fields. For simplicity, we considered the problem in one dimension, treating separately passes from right to left, left to right, and the combined data in order to generate the directional and nondirectional phase fields (*Figure 4A,B*, respectively). We ignored any shifts in place field centers for different running directions (e.g., *Battaglia et al., 2004*; *Huxter et al., 2008*) and assumed that the place cells did not engage in multiple reference frames (*Jackson and Redish, 2007*; *Fenton et al., 2010*).

For the independent coding model, we find that peer prediction provides a higher level of information about a neuron's firing than predictions based on place and nondirectional phase fields, despite the absence of intra-assembly coordination in our simulated data (*Figure 4C*, green and purple). However, prediction based on place fields and directional phase fields outperforms both of these metrics (*Figure 4C*, red). Therefore, previous evidence for intra-assembly coordination can be explained by a failure to account for the phase dependence of CA1 firing. Instead, our analysis

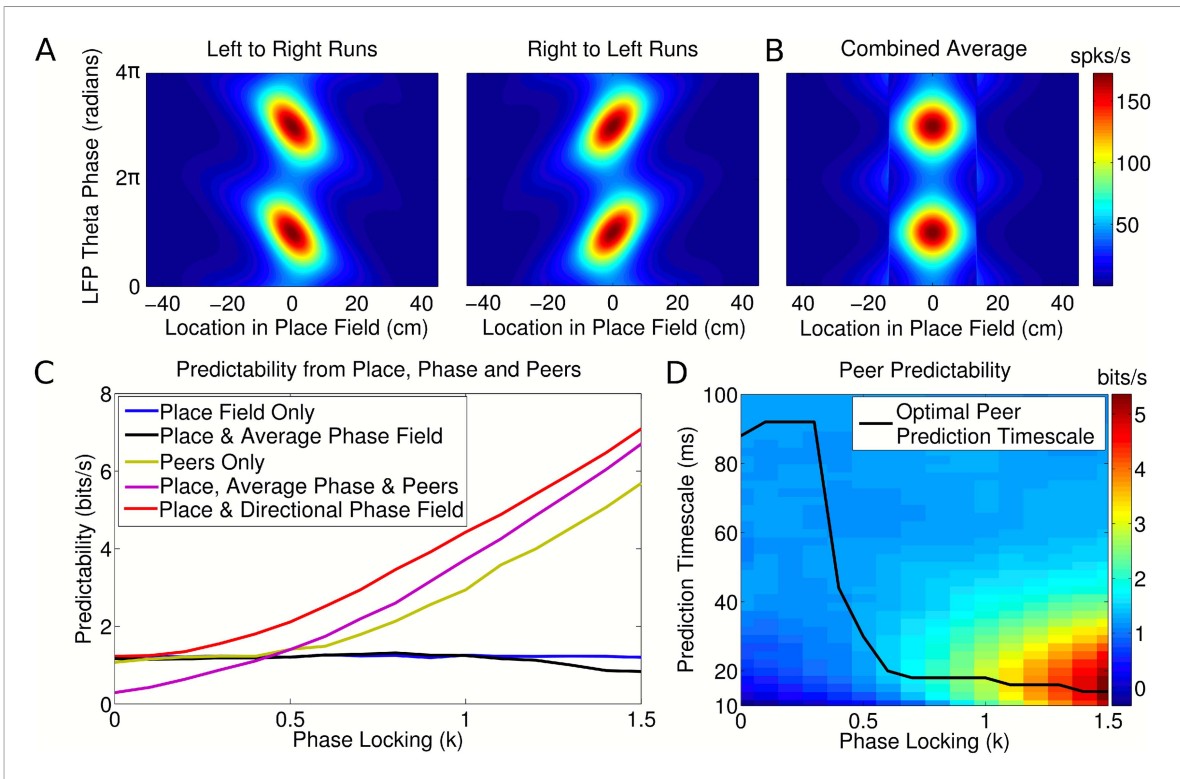

**Figure 4**. Peer prediction analysis for an independent population code. (**A**) Combined place and phase fields constructed from simulated data using only runs with a single direction. (**B**) Place/phase field constructed from a combination of both running directions, as used by *Harris et al. (2003)*. (**C**) Predictability analysis, using various combinations of place, phase, and peer activity. When using the nondirectional phase field of *Harris et al. (2003)*, an additional peer predictability emerges (black vs green and purple). However, this additional predictability is seen to be erroneous if the directional phase field is used to predict activity (red). (**D**) Dependence of peer predictability on the peer prediction timescale and phase locking of individual cells, for an independent population code. The heat map shows the predictability of a cell's activity from peer activity (cf. part **C**, green line). The optimal peer prediction timescale depends on the amount of phase locking. The 20 ms characteristic timescale of peer correlations reflects independent phase precession of single cells rather than transient gamma synchronization of cell assemblies.

The following figure supplements are available for figure 4:

**Figure supplement 1**. Change in information after addition of peer activity to prediction metrics.

**Figure supplement 2**. Results of prediction analysis on individual sessions.

indicates that independent phase precession of CA1 neurons is sufficient to account for observations concerning membership of CA1 assemblies. We also find that nondirectional phase fields (*Figure 4B*), as assumed by (*Harris et al., 2003*), yield little improvement in predictability of a neuron's firing compared with predictions based on the place field alone, and for high phase locking are detrimental (*Figure 4C*, blue vs black). While *Harris et al. (2003)* found that nondirectional phase fields generally do improve prediction, this discrepancy may arise from more complex details of experimental data in open exploration, for example a nonuniform distribution of running directions through the place field, which would cause the information in nondirectional phase fields to increase.

Because peers share a relationship to a common theta activity and implement similar rules for generation of firing, a cell's activity in the independent coding model can nevertheless be predicted from that of its peers in the absence of information about location or theta phase (*Figure 4C*, green). The quality of this prediction is dependent on the timescale at which peer activity is included in the analysis, so that the optimal timescale for peer prediction provides a measure of the temporal resolution of assembly formation. In experimental data the optimal timescale for peer prediction is approximately 20 ms, which corresponds to the gamma rhythm and the membrane time constant of CA1 neurons (*Harris et al., 2003*). We find that in the independent coding model the optimal peer prediction timescale depends strongly on phase locking (*Figure 4D*). Even though the model does not incorporate gamma oscillations or neuronal membrane properties, high values of phase locking also show a striking peak in peer predictability around the 20 ms range (*Figure 4D*). We show below that for running speeds in the range 35–75 cm/s phase locking is likely to lie within the range at which the observed 20 ms prediction timescale dominates. Thus, the 20 ms timescales found both here and experimentally are explainable as a signature of the common, independent phase locking of place cells to the theta rhythm, rather than transient gamma coordination or intrinsic properties of CA1 neurons.

While the above analysis demonstrates that independent coding is consistent with previous experimental results, it does not exclude the presence of coordinated assemblies. In particular, it is not clear whether, when applied to experimental data, including information about peer activity would continue to improve prediction compared to place and directional phase fields alone. We therefore applied the prediction analysis based on directional phase fields to experimental datasets recorded from CA1 place cells (*Mizuseki et al., 2014*). To provide benchmarks for the interpretation of experimental results, we also analyzed simulated datasets generated with either independent coding or coordinated assemblies. We simulated datasets with the same number of sessions and recorded cells per session as the experimental dataset in order to obtain measures of peer prediction performance expected under each hypothesis (see 'Materials and methods'). In simulations of independent cells, we found that information about peer activity continues to improve predictability compared to prediction from place and directional phase fields alone. The source of this predictability was found to be the common modulation of firing rate in each cell with the running speed of the animal, which is a further single cell coding feature not previously accounted for in prediction analyses (*McNaughton et al., 1984*; *Czurko et al., 1999*; *Huxter et al., 2003*; *Ahmed and Mehta, 2012*). We therefore included in our analysis an additional prediction factor, termed the velocity modulation factor (see 'Materials and methods').

After accounting for rate fields, directional phase fields and velocity modulation factors, inclusion of peer information increased the predictability of 84% of place cells simulated through coordinated coding, but only 38% of cells simulated through independent coding (see *Table 1* for a summary of all prediction metrics). On average, information decreased by 0.047 bits/s for each cell simulated by independent coding and increased by 0.24 bits/s for coordinated coding when peer information was added (Wilcoxon signed rank test, $p = 3.9 \times 10^{-17}$ and $p = 9 \times 10^{-83}$, respectively, *Figure 4—figure supplement 1*). Thus, this new prediction analysis which accounts for directional phase fields and velocity modulation can effectively distinguish between independent and coordinated coding.

When we applied this prediction analysis to experimental data, prediction performance improved for 75.7% ($\pm$5.7%, SEM, $n = 10$ sessions) of experimentally observed place cells when phase fields were included and 77.8% ($\pm$3.7%) of place cells when velocity modulation factors were included. In contrast, prediction performance improved for only 32% ($\pm$11%) of the experimentally observed place cells when peer information was included after accounting for single cell coding properties (*Figure 4—figure supplement 2* shows the results for individual experimental sessions). On average, addition of peer information decreased the predictability of each cell by 0.049 bits/s ($\pm$0.013, SEM,

**Table 1.** Performance of prediction metrics on experimental and simulated data

| Prediction metric | Independent coding | Coordinated coding | Experimental data |
|---|---|---|---|
| Location | 100% | 100% | 44.6% (SEM 5.8%) |
| Running speed | 99.3% | 99.7% | 77.8% (SEM 3.7%) |
| Phase field | 99.3% | 100% | 75.7% (SEM 5.7%) |
| Peer activity | 38% | 84.3% | 32.5% (SEM 11%) |

The percentage of cells for which prediction performance increased with the addition of each metric. Percentages refer to the number of cells for which information increased when the specified metric was included in addition to those listed in rows above. Note that for velocity, phase and peer prediction, only those cells for which prediction performance improved with information about location were considered. Simulations demonstrate that, after taking into account place fields, velocity modulation factors and phase fields, information about peer activity improves prediction for the majority of cells when coordination is present, but not when cells are independent. Experimental data are consistent with independent coding.

$n$ = 270 cells, Wilcoxon signed rank test, $p = 1.4 \times 10^{-6}$), in agreement with independent coding simulations and in contrast to coordinated coding simulations. Hence, after fully accounting for the directional properties of phase fields and the dependence of firing rate on running speed, peer prediction analysis supports independent coding as the basis of experimentally observed place cells in CA1. Therefore, based on comparison of simulated with experimental datasets, coordinated assemblies appear unlikely to account for the observed activity in CA1.

## Independent coding accounts for phase sequences

While the above analysis demonstrates that intra-assembly interactions are not required to account for membership of CA1 assemblies, several studies support a role for inter-assembly coordination in the generation of theta sequences (*Dragoi and Buzsáki, 2006*; *Foster and Wilson, 2007*; *Maurer et al., 2012*; *Gupta et al., 2012*). We therefore investigated whether the independent coding or coordinated assembly model would better account for the results of these studies. We focus initially on the path length encoded by spike sequences, which we define as the length of trajectory represented by the sequence of spikes within a single theta cycle. Experimental data show that this path length varies with running speed (*Maurer et al., 2012*; *Gupta et al., 2012*), but it is not clear whether this phenomenon is a feature of independent coding or instead results from coordination between assemblies. To address this we first derived analytical approximations to the sequence path length for strong phase coding, where $k \rightarrow \infty$ (*Supplementary file 1*, Appendix: A2). This analysis predicts a linear increase in sequence path length with running speed, but with a lower gradient than that found experimentally (*Maurer et al., 2012*). Hence, independent coding with strong phase locking does not quantitatively explain the changes in sequence properties with running speed.

We reasoned that independent coding might still explain observed sequence path lengths if a more realistic tradeoff between rate and phase coding is taken into account. To test this, we varied phase locking $k$ and decoded the path length following the method of *Maurer et al. (2012)*, which decodes the location represented by the population at each time bin in a theta cycle to estimate the encoded trajectory. We found that a good match to the data of *Maurer et al. (2012)* can be obtained by assuming that the degree of phase locking increases with running speed (*Figure 5A*). This is due to the dependence of the decoded path length on the strength of phase locking (*Figure 5—figure supplement 1A*).

*Maurer et al. (2012)* found that the compression factor $c$, which measures the compression of an encoded trajectory into a single theta cycle, also depends on running speed. To test whether independent coding might account for this observation, we investigated the behavior of the fast and slow slopes of population activity (as shown in *Figure 3B*), representing assembly propagation at theta timescales and behavioral timescales, respectively (i.e., $v_p$ and $v$). In the analysis of *Maurer et al. (2012)*, the compression factor was estimated as the ratio of these two quantities. Following again the methods used by *Maurer et al. (2012)* to decode the fast and slow slopes from spiking data, we found that the dependence of the decoded fast slope on running speed in our simulated data matches experimental data provided that phase locking is again made dependent on running speed

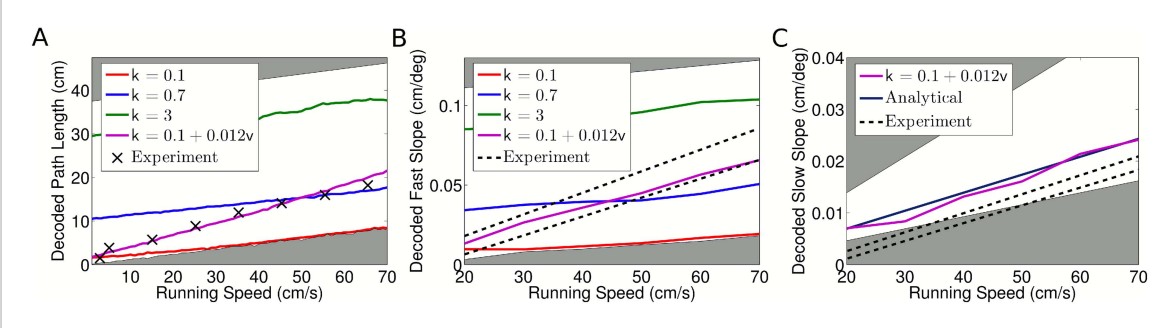

Figure 5. Decoded sequence path lengths and population activity propagation speeds. (A) With constant phase locking, the decoded path length increases linearly with running speed, but to account for experimental data a dependence of phase locking on running speed is required. The shaded regions show lower and upper bounds ($k = 0$ and $k = \infty$). (B) Dependence of decoded fast slope on running speed (cf. our *Figure 3B*; Figure 3 of *Maurer et al. (2012)*). Again, a match to the data requires a velocity dependent phase locking. (C) The decoded slow slope matches the analytical value, where the population travels at the running speed *v*. Bounds show LFP theta frequencies below 4 Hz (upper bound) and above 12 Hz (lower bound) at each given running speed.

The following figure supplements are available for figure 5:

**Figure supplement 1**. Dependence of decoded sequence path lengths, fast slopes, and slow slopes on phase locking.

**Figure supplement 2**. Results of shuffling analysis.

(*Figure 5B*, *Figure 5—figure supplement 1B*). However, the slower behavioral timescale dynamics did not match those reported by *Maurer et al. (2012)*. Our decoded values for the slow slope closely matched the true value based on the rat's running speed. In contrast, the values reported by *Maurer et al. (2012)* are considerably lower (*Figure 5C*) which, if correct, would suggest that the population consistently moved more slowly than the rat, even moving backwards while the animal remained still. Because of this discrepancy we could not reproduce the compression factor reported by *Maurer et al. (2012)*. Nevertheless, the independent coding model accurately reproduces the theta timescale activity reported by *Maurer et al. (2012)*.

The above analysis has two important implications. First, both the decoded sequence path length and theta-compressed propagation speed in the independent coding model match experimental data provided the degree of theta modulation of spike output increases linearly with running speed. This dependence of phase locking on running speed is consistent with the observed increase in LFP theta amplitude (*McFarland et al., 1975*; *Maurer et al., 2005*; *Patel et al., 2012*), and is a novel prediction made by our model. Second, since the temporal delays between cells are determined by the propagation speed $v_p$, the close match of this quantity to experimental data confirms the dependence of temporal delays on running speed predicted by our model, and argues against models based on fixed delays (*Diba and Buzsáki, 2008*; *Geisler et al., 2010*).

Further experimental support for the notion of inter-assembly coordination has come from an analysis suggesting that single cell phase precession is less precise than observed theta sequences (*Foster and Wilson, 2007*). This conclusion relies on a shuffling procedure which preserves the statistics of single cell phase precession yet reduces intra-sequence correlations. However, performing the same shuffling analysis on data generated by our independent coding model also reduced sequence correlations (t-test, $p < 10^{-46}$) (*Figure 5—figure supplement 2*). The effect arises because the shuffling procedure does not preserve the temporal structure of single cell phase precession, despite preserving the phasic structure (*Figure 5—figure supplement 2A,B*). Hence, the phase–position correlations are unaffected, while the time–position correlations and hence sequence correlations are disrupted (*Figure 5—figure supplement 2C,D*). Thus, inter-assembly coordination is not required to account for these findings.

Nevertheless, although these results are reproducible by the independent coding model, it remains possible that coordinated assemblies underly the observed theta sequences. In particular, it is unclear whether this shuffling procedure could be modified to obtain a test for assembly coordination

with greater statistical specificity and if so, whether it would reveal assembly coordination within experimental datasets. To address these questions, we analyzed experimental data along with data generated by independent coding and coordinated assembly models, using a modified version of this shuffling procedure (see 'Materials and methods'). We found that the new shuffling procedure successfully detected assembly coordination with a statistical power of 81% (calculated for datasets containing the same number of sessions, cells, and sequences as our experimental dataset). When applied to experimental data from CA1, the shuffling test failed to detect any significant effect of shuffling (t-test, p = 0.28, 2436 events), as in most (74%) of the simulated independent coding datasets (*Figure 5—figure supplement 2E,F*). This failure to detect evidence of assembly coordination gives further support to the independent coding hypothesis.

In additional support for the coordinated assembly hypothesis, *Dragoi and Buzsáki (2006)* performed an analysis suggesting that, during continuous locomotion around a rectangular track, some cell pairs show a lap by lap covariation of firing rates (termed the dependent pairs). These cell pairs were found to spike with a more reliable temporal lag within theta cycles than cell pairs whose firing rates are independent, which was interpreted as evidence for direct interactions between dependent neurons. To test whether these results are instead consistent with independent coding, we applied the analysis of lap by lap firing rate covariations to data from simulations of the independent coding model. We found a similar fraction of apparently dependent cell pairs to that reported by *Dragoi and Buzsáki (2006)*, despite the absence of any true dependencies between cells in the model (see 'Materials and methods'). Hence, this analysis artificially separates homogeneous populations of place cells into apparently dependent and independent cell pairs. Moreover, these dependent and independent cell groups displayed different spatial distributions of place fields, with dependent cell pairs generally occuring closer together on the track (Wilcoxon rank sum test, $p = 1.8 \times 10^{-16}$). By separating a homogeneous population of cells into dependent and independent groups, the analysis therefore introduces a sampling bias, leading to dependent cell pairs having different properties. While we were unable to reproduce the analysis of the temporal lags in each group due to a lack of information provided within the original study (see 'Materials and methods'), the emergence of dependent cell pairs with measurably different properties in independent coding simulations nevertheless demonstrates that these results are not indicative of interactions between neurons.

Finally, precise coordination of theta sequences has been suggested on the basis that theta sequence properties vary according to environmental features such as landmarks and behavioral factors such as acceleration, with sequences sometimes representing locations further ahead or behind the animal (*Gupta et al., 2012*). To establish whether independent coding could also account for these results, we generated data from our model and applied the sequence identification and decoding analysis reported by *Gupta et al. (2012)*. We found that, even for simulated data based on pure rate coding with no theta modulation ($k = 0$), large numbers of significant sequences were detected at high running speeds (*Figure 6A*). Therefore, to test the performance of the full sequence detection and Bayesian decoding protocol used by (*Gupta et al., 2012*), we analyzed two simulated datasets—one with a realistic value of phase locking ($k = 0.5$, *Figure 6B–D*, solid lines) and another with zero phase locking (i.e., no theta related activity, *Figure 6B–D*, dashed lines). In both cases, applying the reported Bayesian decoding analysis yielded similar decoded path lengths to those found experimentally (*Figure 6C,D*). Importantly, there was an inverse relationship between the ahead and behind lengths decoded from the simulated data, which reproduces the apparent shift in sequences ahead or behind the animal observed in experimental data (cf. Figure 4c of *Gupta et al. (2012)*). This effect was dependent on the density of recorded place fields on the track and the threshold for the minimum number of cells in a theta cycle required for sequence selection (*Figure 6—figure supplement 1*). As these results were obtained both in the case with realistic phase coding and in the case with only rate coding (and therefore no theta sequences), the properties of the decoded trajectories are not related to theta activity within the population. Hence, these data do not constrain models of theta activity in CA1.

In total, our analysis demonstrates that a traveling wave model based on independent phase coding for CA1 theta states is consistent with existing experimental data. Thus, neither intra- nor inter-assembly interactions are required to explain spike sequences observed in CA1 during theta states. Our analyses of experimental data along with simulations from each hypothesis render it unlikely that assembly coordination significantly shapes the structure of theta sequences or CA1 cell assemblies.

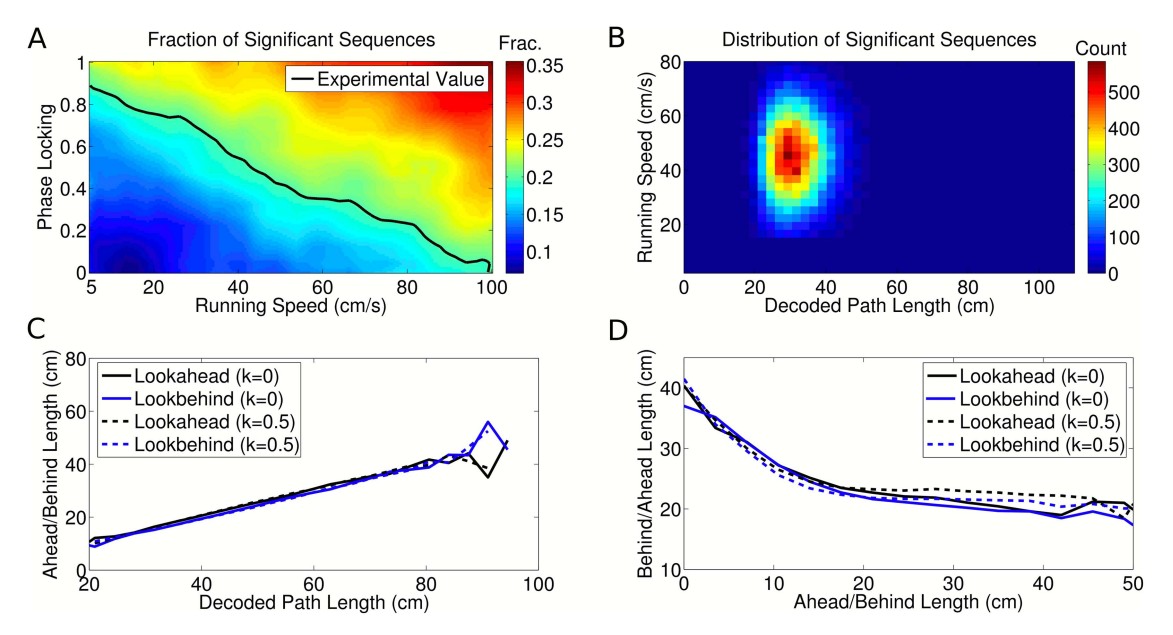

**Figure 6**. Analysis of individual sequence statistics. (**A**) The fraction of theta cycles which are classified as 'significant sequences' according to the **Gupta et al. (2012)** analysis, as a function of running speed and phase locking (for simulated data generated under the independent coding model). Large fractions of significant sequences are generated even without phase coding or theta sequences within the population (i.e., at $k = 0$). The black line shows the fraction reported experimentally. (**B**) The distribution of significant sequences over running speed and decoded path length for simulated data with phase locking $k = 0.5$, as calculated by **Gupta et al. (2012)** (cf. their Figure 1c). (**C**) The relationship between decoded path length and decoded ahead and behind lengths for significant sequences, calculated for a dataset with no theta activity ($k = 0$) and a dataset with realistic theta activity ($k = 0.5$). (**D**) The relationship between the ahead length of the sequence and the behind length of the sequence for these two datasets. Note that the properties of the decoded trajectories do not depend on the theta activity in the data. This replicates the experimental data (cf. Figure 4a-c of **Gupta et al. (2012)**), showing that similar trajectories are decoded by this algorithm regardless of the presence of theta sequences.

The following figure supplement is available for figure 6:

**Figure supplement 1**. Dependence of decoded trajectories on the number of cells in a sequence.

Below, we investigate some functional consequences of the independent coding and coordinated assembly hypotheses and show that, despite the advantage of assembly coordination in generating robust sequential activity patterns, it suffers from severe limitations in remapping and storage of multiple spatial maps. Independent coding offers a solution to this problem, allowing flexible generation of sequential activity over multiple spatial representations.

## Linear phase coding constrains global remapping

What are the advantages of independent coding compared to sequence generation through interactions between cell assemblies? When an animal is moved between environments, the relative locations at which place cells in CA1 fire remap independently of one another (e.g., **O'Keefe and Conway, 1978**; **Wilson and McNaughton, 1993**). This global remapping of spatial representations poses a challenge for generation of theta sequences through coordinated assemblies as synaptic interactions that promote formation of sequences in one environment would be expected to interfere with sequences in a second environment. Indeed, in the coordinated assembly model, simulations of remapping reduced single cell phase precession to below the level of independent cells (i.e., of an identical simulation with interactions between cells removed). Remapping in the coordinated coding model also substantially reduced firing rate and population oscillations (**Figure 7—figure supplement 1**). This decrease in firing rate following remapping contradicts experimental data showing an increase in firing rate in novel environments (**Karlsson and Frank, 2008**). It is not immediately clear whether the independent coding model faces similar constraints on sequence generation across different spatial

representations. We therefore addressed the feasibility of maintaining theta sequences following remapping given the assumptions that underpin our independent coding model.

We first consider the possibility that following remapping the phase lags between cell pairs remain fixed—that is, while two cells may be assigned new firing rate fields, their relative spike timing within a theta cycle does not change. This scenario would occur if the phase lags associated with phase precession were generated by intrinsic network architectures (e.g., *Diba and Buzsáki, 2008*; *Geisler et al., 2010*; *Dragoi and Tonegawa, 2011*, *2013*) or upstream pacemaker inputs. For fixed phase lags, place cells display linear phase coding, whereby a cell continues to precess in phase outside of its rate coded firing field at a constant rate (*Figure 7A*). In this scenario, the phase lag between two neurons depends linearly on the distance between their place field centers, while cells separated by multiples of a place field width share the same phase (*Figure 7A*). Each cell pair

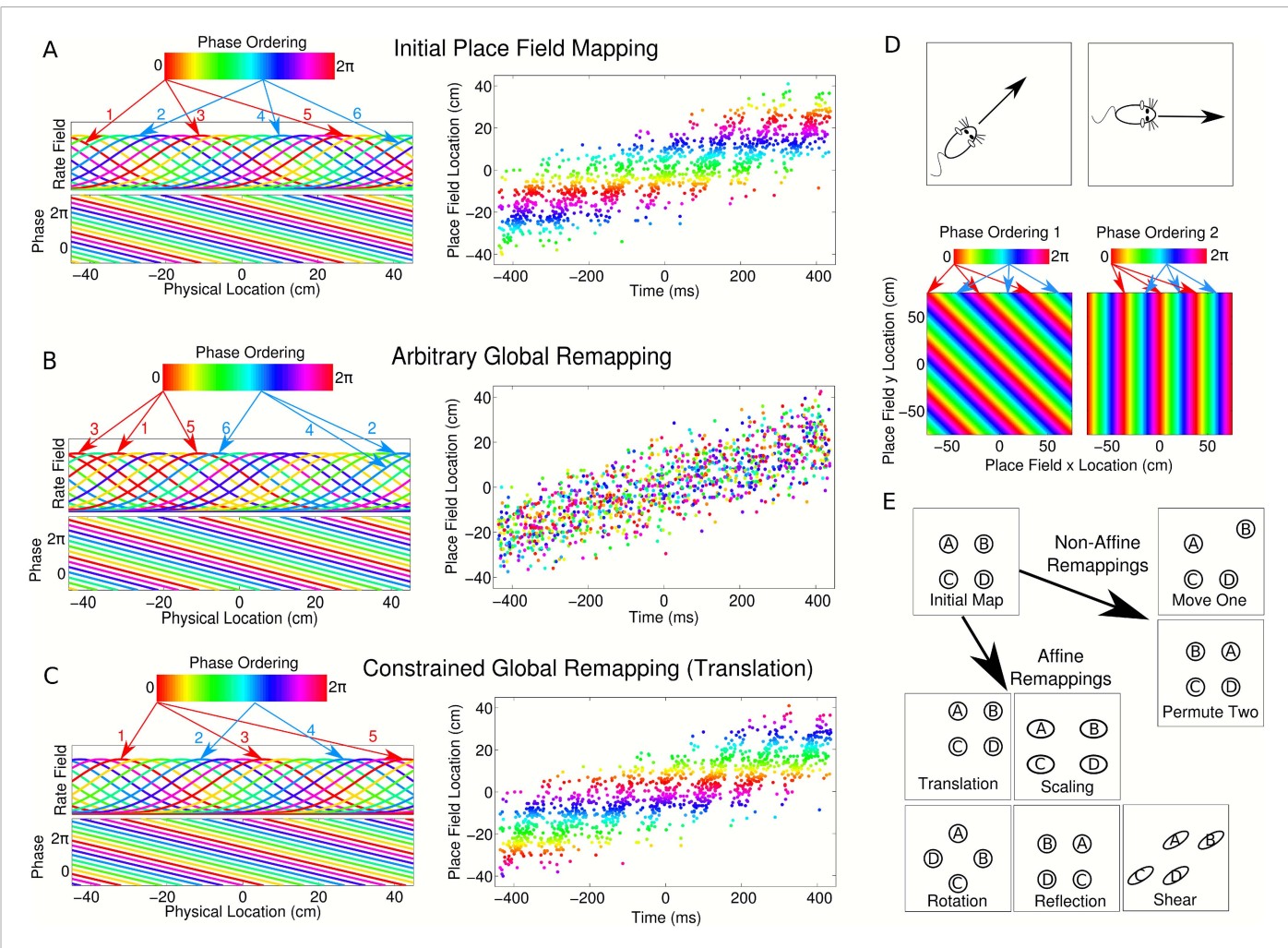

**Figure 7**. Properties of CA1 populations governed by linear phase coding. (**A**) On a linear track, cells which precess linearly in phase maintain fixed theta phase lags. This is illustrated as a phase ordering (colored bar), which describes the relative phase of each cell (arrows show locations of cells at each phase). Each cell has a constant, running speed dependent frequency and a fixed phase offset to each other cell. (**B**) A complete global remapping with phase lags between cells held fixed. Theta sequences and population oscillations are abolished. (**C**) In a constrained place field remapping, theta sequences are preserved. (**D**) In open environments, phase lags depend on running direction. The set of population phase lag configurations needed to generate sequences in each direction is called a phase chart. (**E**) If a population has a fixed phase chart, the possible remappings are restricted to affine transformations.

The following figure supplement is available for figure 7:

**Figure supplement 1**. Remapping with coordinated assemblies.

therefore has a fixed phase lag in all environments and all cells can in principle be mapped onto a single chart describing their phase ordering (*Figure 7A*). If this mechanism for determining phase ordering is hardwired, then following arbitrary global remapping, cells with nearby place field locations will in most cases no longer share similar phases (*Figure 7B*). As a result, theta sequences and the global population theta will in general be abolished (*Figure 7B*). However, there exist a limited set of remappings which in this scenario do not disrupt the sequential structure of the population (e.g., *Figure 7C*). On a linear track, these remappings are: translation of all place fields by a fixed amount, scaling of all place fields by a fixed amount, and permuting the place field locations of any cell pair with zero phase lag.

When considering global remapping in an open environment similar constraints apply. Because the phase lag between any two cells depends on running direction (e.g., *Huxter et al., 2008*), the population phase ordering must always be aligned with the direction of movement (*Figure 7D*). Hence, in open environments, the notion of a phase chart must be extended to include a fixed phase ordering for each running direction. Given such a fixed phase chart, a set of remappings known as *affine transformations* preserve the correct theta dynamics (see *Supplementary file 1*, Appendix: A7). Such remappings consist of combinations of linear transformations (scaling, shear, rotation, and reflection) and translations (*Figure 7E*). Remappings based on permutation of place field locations of synchronous cells, which are permissible in one dimensional environments, are no longer tenable in the two dimensional case due to constraints over each running direction.

Since place cell ensembles support statistically complete (i.e., non-affine) remappings (e.g., *O'Keefe and Conway, 1978*) while maintaining phase precession, CA1 network dynamics are not consistent with the model outlined above. Moreover, this analysis demonstrates that previous models based on fixed temporal delays between cells (e.g., *Diba and Buzsáki, 2008*; *Geisler et al., 2010*) cannot maintain theta sequences following global remapping. Nevertheless, it remains possible that CA1 theta dynamics are based on fixed phase charts, provided that multiple such phase charts are available to the network, similar to the multiple attractor charts which have been suggested to support remapping of firing rate (*Samsonovich and McNaughton, 1997*). In this case, each complete remapping recruits a different phase chart, fixing a new set of phase lags in the population. The number of possible global remappings that maintain theta sequences is then determined by the number of available phase charts. Such a possibility is consistent with recent suggestions of fixed sequential architectures (*Dragoi and Tonegawa, 2011*, *2013*) and has not been ruled out in CA1. It is also of interest that affine transformations are consistent with the observed remapping properties in grid modules (*Fyhn et al., 2007*), suggesting that a single phase chart might be associated with each grid module.

## Sigmoidal phase coding enables theta sequence generation and flexible global remapping

The above analysis demonstrates that both coordination of assemblies and independent, linear phase coding pose severe restrictions on global remapping which appear at odds with experimental observations. We asked if it is possible to overcome these constraints so that phase sequences can be flexibly generated across multiple environments. We reasoned that experimental data on phase precession only imply that phase precesses within a cell's firing field and need not constrain a cell's phase outside of its firing field. We therefore implemented a version of the independent coding model in which firing phase has a sigmoidal relationship with location (*Figure 8A–B*, solid line; *Supplementary file 1*, Appendix: A5), such that phase precesses within the firing field but not outside of the field. In this case, each cell's intrinsic frequency increases as the animal enters the spatial firing field and drops back to LFP frequency when the animal exits the firing field (*Figure 8C*, solid line). This is in contrast to the linear phase model and previous work with fixed delays (*Geisler et al., 2010*) in which each cell's intrinsic frequency is always faster than the population oscillation, both inside and outside of the place field (*Figure 8C*, dashed line). In a given environment, spike phase precession and sequence generation in a population of cells with sigmoidal phase coding (*Figure 8D–F*) are similar to models in which cells have linear phase coding. However, in addition, sigmoidal phase coding enables theta sequences to be generated after any arbitrary global remapping (*Figure 8G*). This flexible global remapping is in contrast to the scrambling of theta sequences following global remapping when cells have linear phase coding (*Figure 8G*). Thus, independent sigmoidal coding is able to account for CA1 population activity before and after global remapping.

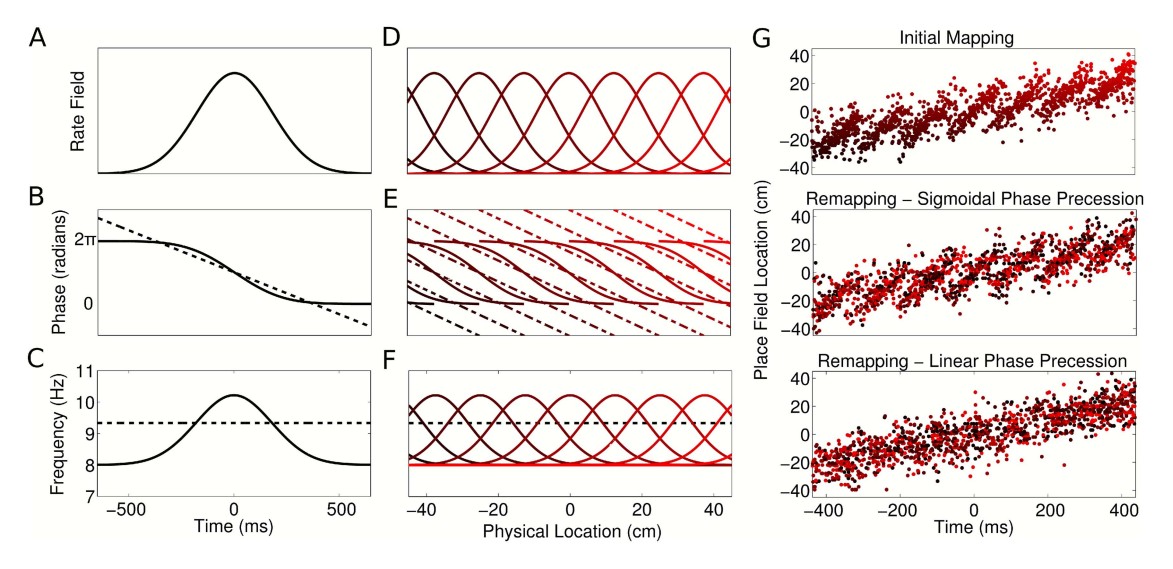

**Figure 8**. Properties of CA1 populations governed by sigmoidal phase coding. (**A**–**C**) Firing rate and intracellular phase and frequency in the linear (dashed lines) and sigmoidal models (solid lines) during the crossing of a place field. In the sigmoidal model, phase precession is initiated inside the place field by an elevation of intracellular frequency from baseline. (**D**–**F**) Firing rate and intracellular phase and frequency for a place cell population on a linear track. In the sigmoidal model, an intracellular theta phase lag between cell pairs develops as the animal moves through their place fields. Outside their place fields, cell pairs are synchronized. (**G**) Global remapping in the linear and sigmoidal models. The sigmoidal model allows arbitrary remapping without disrupting population sequences.

Linear and sigmoidal models of phase coding lead to distinct experimentally testable predictions. Recordings of the membrane potential of CA1 neurons in behaving animals show that although spikes precess against the LFP, they always occur around the peak of a cell's intrinsic membrane potential oscillation (MPO) (*Harvey et al., 2009*). Therefore the intrinsic phase $\phi$ of each cell in our model (*Figure 2D,E*) can be interpreted as MPO phase. While data concerning the MPO phase outside of the firing field are limited, such data would likely distinguish generation of theta sequences based on a linear and sigmoidal phase coding. If CA1 implements linear phase coding, then the MPO of each cell should precess linearly in time against LFP theta at a fixed (velocity dependent) frequency, both when the animal is inside the place field and when the animal is at locations where the cell is silent (*Figure 8A–C*, dashed line). Alternatively, sigmoidal phase coding predicts that precession of the MPO against the LFP occurs only inside the firing rate field (*Figure 8A,B*, solid line) and that the MPO drops back to the LFP frequency outside of the place field (*Figure 8C*, solid line) as reported by *Harvey et al. (2009)*. A further prediction of sigmoidal coding is that, in contrast to models based on fixed delays (*Diba and Buzsáki, 2008*; *Geisler et al., 2010*), the phase lag between any two cells changes when the animal moves through their place fields. Outside their place fields the cells are synchronized with each other and with the LFP, whereas a dynamically shifting phase lag develops as the animal crosses the place fields (*Video 2*). Finally, phase precession under the sigmoidal model behaves differently to the linear model in open environments. In the linear model, the phase chart fixes a different population phase ordering for each running direction, so that spike phase depends on the location of the animal and the instantaneous direction of movement. In the sigmoidal model, however, each cell has a location dependent frequency, so that the spike phase depends on the complete trajectory through the place field and no explicit directional information is required (see *Supplementary file 1*, Appendix: A6). Rather, the directional property of the sequence arises purely through a location dependent oscillation frequency in each cell combined with the trajectory of the animal through each place field. In summary, our analysis demonstrates how evaluation of theta sequences following global remapping and of theta phase within and outside of a cell's firing field will be critical for distinguishing models of CA1 assemblies and theta generation.

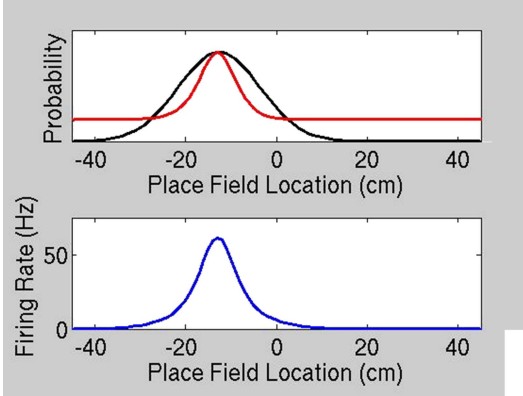

**Video 2.** Population dynamics with sigmoidal phase coding. Top: Distribution of the rate (black) and phasic (red) tuning curves for a population of sigmoidal phase coding place cells during constant speed locomotion on a linear track. Bottom: The evolution of the overall firing rate distribution in the population. Again, the population firing rate undergoes oscillations at LFP theta frequency and the center of mass of the population activity shifts from behind the animal to ahead of the animal in each theta cycle. However, in this case cells with place field centers distant from the animal's current location are synchronized with zero phase lag.

## Discussion

Our analysis demonstrates how complex and highly structured population sequences can be generated without coordination between neurons. In contrast to previous suggestions (*Harris et al., 2003*; *Dragoi and Buzsáki, 2006*; *Foster and Wilson, 2007*; *Maurer et al., 2012*; *Gupta et al., 2012*), we find that the theta-scale population activity observed in CA1 is consistent with phase precession in independent cells, without interactions within or between cell assemblies. We demonstrate that independent coding enables flexible remapping of CA1 population activity while maintaining the ability to generate theta sequences. These properties are consistent with maximization of the capacity of CA1 for representation of distinct spatial experiences.

The independent coding hypothesis leads to a novel view of the CA1 population as a fast moving traveling wave with a slower modulatory envelope. This model implements an invariant phase code via a change in the frequency and temporal delay between cells with running speed. Amplitude modulation of the envelope provides a mechanism for multiplexing spatial with nonspatial information, such as task specific memory items (*Wood et al., 2000*) and sensory inputs (*Rennó-Costa et al., 2010*). The independence of each neuron naturally explains the robustness of phase precession against intrahippocampal perturbations (*Zugaro et al., 2005*), an observation which is difficult to reconcile with models based on assembly interactions. Depending on the exact nature of the single cell phase code, independent phase coding can enable theta sequences to be maintained with arbitrary global remapping. This flexibility may maximize the number and diversity of spatial representations that CA1 can provide to downstream structures, offering a strong functional advantage over mechanisms based on interactions between cell assemblies.

Independent phase coding leads to new and experimentally testable predictions that distinguish mechanisms of CA1 function during theta states. First, an absence of coordination within or between assemblies has the advantage that neural interactions do not interfere with sequence generation after global remapping. Rather, for independent coding models the constraints on sequence generation following remapping arise from the nature of the phase code. With linear phase coding the set of sequences available to the network is fixed, resulting in a limited set of place field configurations with a particular mathematical structure (*Figure 7*). Interestingly, the remappings observed in grid modules (*Fyhn et al., 2007*), but not CA1, are consistent with those predicted for networks with a single fixed set of theta phase lags called a phase chart. These findings, together with the fact that the temporal delays between cells depend on running speed, argue against previous models based on fixed delays within CA1 populations (*Diba and Buzsáki, 2008*; *Geisler et al., 2010*). Nevertheless, more complex scenarios with multiple phase charts could explain CA1 population activity during theta oscillations and 'preplay', which suggests a limited remapping capacity for CA1 (*Dragoi and Tonegawa, 2011*, *2013*). Alternatively, sigmoidal phase coding massively increases the flexibility for global remapping as cells can remap arbitrarily while maintaining coherent theta sequences within each spatial representation (*Figure 8*). Second, linear and sigmoidal phase coding predict distinct MPO dynamics. With linear phase coding, the temporal frequency of each MPO is independent of the animal's location. With sigmoidal phase coding, the MPO frequency increases inside the place field, so that phase precession occurs inside but not outside the place field. In this case, only the spiking assembly behaves as a traveling wave, whereas the MPOs of cells with place fields distant from the animal are

phase locked to the LFP. Sigmoidal phase precession could emerge due to inputs from upstream structures (*Chance, 2012*) or be generated intrinsically in CA1 place cells (*Leung, 2011*). Finally, in contrast to linear phase coding populations, sigmoidal phase coding populations do not require additional information from head direction or velocity cells to generate directed theta sequences in open environments. Instead, sigmoidal theta sequences are determined solely by the recent trajectory of the rat through the set of place fields together with a location dependent oscillation frequency, consistent with recent observations of reversed theta sequences during backwards travel (*Cei et al., 2014*; *Maurer et al., 2014*). In summary therefore, these two models may be distinguished experimentally on the basis of observations of the number of non-affine remappings in CA1, the intracellular frequency and delay between place cells as a function of location and of the dependence of firing phase on the trajectory through a place field in open environments.

While theta sequences of CA1 activity are most commonly observed during spatial navigation, similar activity patterns associated with short term memory have been observed during wheel running (*Pastalkova et al., 2008*). In this situation each cell's activity depends on the phase of the LFP theta rhythm and on the temporal location within an 'episode field' rather than a place field. Our model can be applied equally well to these internally generated sequences if the rate coded episode field is assumed to have a similar temporal structure to a place field. An entirely different class of sequences, however, is observed during non-theta states such as sharp wave ripples (SWR) (*Buzsáki et al., 1992*; *Diba and Buzsáki, 2007*). In contrast to theta sequences, SWR sequences are generally observed during states of immobility and are believed to arise from synchronous discharge in CA3 (*Buzsáki et al., 1983*). Because SWR sequences are generated without co-occurence of longer time-scale firing fields or theta oscillations, they cannot be accounted for by the independent coding schemes that we investigate here, in which rate and phase information determine the activity of each cell. Instead, the nature of cell assemblies in CA1 may be highly state dependent, operating in at least two modes. During theta states, sequences are generated by independently precessing neurons, whereas during SWRs sequences may result from interactions between consecutively activated cell assemblies.

Can independent coding account for manipulations that modify place cell dynamics? Administration of cannabinoids disrupts phase precession by CA1 neurons and impairs spatial memory, but does not appear to affect the rate coded place firing fields of CA1 neurons (*Robbe and Buzsáki, 2009*). This dissociation between rate and phase coding can be accounted for in our model by assuming that rate fields are maintained while phase fields are disrupted (*Figure 2A*) or the degree of phase locking ($k$) is substantially reduced (*Figure 2B*). In contrast, increased in-field firing of place cells following optogenetic inactivation of hippocampal interneurons (*Royer et al., 2012*) can be accounted for in our model by increased $N_{spikes}$, while altered phase of place cell firing following inactivation of parvalbumin interneurons can be accounted for in our model by modifying the phase fields (*Figure 2A*) of the place cells. Important future tests of the independent coding model will include comparison of its predictions of sequence activity, remapping and intracellular dynamics to experimental measures made during these kinds of manipulations.

Our independent coding model offers a comprehensive account of population activity in CA1 during theta states and makes new predictions for coordination of network dynamics and remapping at the population level, but it does not aim to distinguish cellular mechanisms for phase precession. Nevertheless, by demonstrating that existing observations of population sequences can be explained by independent coding our model argues against mechanisms for phase precession that rely on synaptic coordination at theta time scales (e.g., *Tsodyks et al., 1996*; *Maurer and McNaughton, 2007*; *Lisman and Redish, 2009*). In contrast, our model does not distinguish between specific single cell mechanisms for phase precession such as dual oscillators (*Lengyel et al., 2003*; *Burgess et al., 2007*), depolarizing ramps (*Mehta et al., 2002*), intrinsic membrane currents (*Leung, 2011*) or dual inputs from CA3 and entorhinal cortex (*Chance, 2012*). Our model is also consistent with inheritance of phase precession in CA1 from upstream circuits in CA3 and entorhinal cortex (*Jaramillo et al., 2014*). However, it argues against the possibility that CA1 inherits coordinated sequences from CA3 (*Jaramillo et al., 2014*). It is possible that CA3 nevertheless generates sequences by synaptic coordination. Because CA3 neurons are connected by dense recurrent collaterals (*Miles and Wong, 1986*; *Le Duigou et al., 2014*), there are likely to be substantial correlations in their output to CA1, which could induce deviations from the independent population code outlined here. However, feedback inhibition motifs such as those found in CA1 may counteract such correlations (*Renart et al., 2010*; *Tetzlaff et al., 2012*; *Bernacchia and Wang, 2013*; *King et al., 2013*; *Sippy and Yuste, 2013*).

Hence, the local inhibitory circuitry in CA1 may actively remove correlations present in its input in order to generate an independent population code (*Ecker et al., 2010*).

A major advantage of independently precessing cell populations is that they provide a highly readable, robust, and information rich code for working and episodic memory in downstream neocortex. In particular, a downstream decoder with access to an independent population code need only extract the stereotyped correlational patterns associated with traveling waves under a given place field mapping. In this way it can flexibly decode activity across a large number of spatial representations. Decoding in the presence of additional correlations would likely lead to a loss of information (*Zohary et al., 1994*). While this loss can to some extent be limited by including knowledge of these additional correlations (*Nirenberg and Latham, 2003*; *Eyherabide and Samengo, 2013*), this likely requires a high level of specificity and therefore a lack of flexibility in the decoder. The flexibility afforded by an independent population code may therefore provide an optimal format for the representation and storage of the vast number of spatial experiences and associations required to inform decision making and guide behavior.

## Materials and methods

### Simulations of CA1 population activity

In the independent coding model, we simulated data from a population of place cells with place field centers $x_c$ and width $\sigma$ which precess linearly through a phase range of $\Delta\phi$ over a distance $2R$ on a linear track using Equation (A3.6) in *Supplementary File 1*. The initial phase $\psi_s$ was either taken as 0, or a uniform random variable $\psi_s \in [0,2\pi]$ set at the beginning of each run. In all simulations, parameters were set as: $2R = 37.5$ cm (*Maurer et al., 2006*), $\Delta\phi = 2\pi$, $\sigma = 9$ cm, $f_\theta = 8$ Hz, $N_{\text{spikes}} = 15$. Finite numbers of place cells were simulated with place field centers $x_c$ which were either uniformly distributed along a linear track with equal spacing or randomly sampled from a uniform distribution over the track. All cells were therefore identical up to a shift in place field center $x_c$. Simulations were performed using Matlab 2010b and 2013b.

Simulations of population activity generated through coordinated assemblies used equations (A4.1–4.5) in *Supplementary File 1*, with the single cell properties simulated as for the independent coding model. The peer interaction timescale was set to $\tau = 25$ ms, and the interaction length for asymmetric excitation was set to $\ell = 10$ cm with an excitatory amplitude of $w_E = 1/4$. The amplitude of the inhibitory weights was varied until the same number of spikes were generated as in the independent coding simulation (for the parameters used in these simulations, the inhibitory amplitude was $w_I = 1/18$).

### Experimental datasets

We used data recorded from CA1 during navigation along a linear track. For details of experimental data see *Mizuseki et al. (2014)*. For the analysis performed in this study, simultaneous recordings of a large number of coactive cells in CA1 are required, which restricted the number of suitable datasets. In particular, we used datasets *ec016.233*, *ec016.234*, *ec016.269*, *ec014.468*, *ec014.639*.

### Prediction analyses

To replicate the results of *Harris et al. (2003)*, we simulated constant speed movement along a linear track, with lap by lap running speeds drawn from a normal distribution with mean 35 cm/s and standard deviation of 15 cm/s. We simulated motion in each direction, using the same set of place fields in each case. We estimated the preferred firing phase at each location from the simulated data using the methods stated in *Harris et al. (2003)*, using either single-direction data or data consisting of runs in both directions to generate nondirectional or directional phase fields. The prediction analysis was performed according to the methods given in *Harris et al. (2003)*. For these initial simulations (*Figure 4*), we used the simulated value of phase locking rather than estimating it from the data. To display the peer prediction performance shown in *Figure 4C*, the optimal prediction timescale for each phase locking value was chosen. This was done separately for the peer only case and the peer plus phase field case.

We then performed additional, more detailed simulations to test the performance of simulated and experimental data when using the new directional phase fields. We separated datasets according to the running direction along a linear track, analyzing each direction individually. In addition to fitting

the place field, phase field, and peer factor used by *Harris et al. (2003)*, we also fitted a velocity modulation factor given by:

$$A(v) = \frac{\sum_t n_t w\left(|v - v_t|\right)}{\sum_t r_0\left(x_t\right) dt w\left(|v - v_t|\right)}, \tag{5}$$

which estimates the changes in firing rate of a cell according to running speed. In the above equation, the notation follows that of *Harris et al. (2003)* (their Supplementary Information), that is, $w$ is a Gaussian smoothing kernel of width 3.5 cm/s, $n_t$ is the number of spikes fired by the cell in time bin $t$, $r_0$ is the estimated firing rate field at location $x$, $x_t$ is the animal's location in time bin $t$, and $v_t$ is its velocity. Our simulations showed that, using the methods of *Harris et al. (2003)*, the phase locking parameter $k$ was underestimated outside of the place field center. Misestimation of phase field parameters introduces false peer predictability in simulated datasets. We therefore replaced their location dependent estimation with a fixed value equal to the phase locking estimated in regions where the place field is over 2/3 its maximum value. We also found that the 5 cm spatial smoothing kernel used by *Harris et al. (2003)* resulted in a high level of spurious peer prediction in simulations based on independent coding, since it extended the boundaries of place fields, allowing non-overlapping peer cells to compensate via inhibitory weights. A smaller kernel of 3.5 cm reduced the rate of false positive for peer prediction and was therefore used instead. We simulated 300 cells in each session of which we randomly sampled 15 for analysis in order to match the number of place cells typically recorded experimentally. 28 laps were simulated for each session and 10 sessions were simulated in total (representing the two running directions over the five experimental sessions we analyzed). Peer prediction was performed at a timescale of 25 ms (the optimal timescale in *Harris et al. (2003)*).

## Changes in sequence properties with running speed

To compare the sequence path length in spiking data generated from the independent coding model to experimental data, we followed the decoding methods outlined in *Maurer et al. (2012)*. Briefly, this involves constructing trial averaged time by space population activity matrices in order to decode the location represented by the population in each time bin over an average theta cycle. The decoded path length is measured as the largest distance between decoded locations within the theta cycle. To test the influence of phase locking in this analysis, $k$ was varied incrementally from 0 to 6 and the sequence path length for the resulting data was calculated in each case. We used the same spatial and temporal bins (0.7 cm and 20° of LFP $\theta$) as the original study.

To calculate the fast and slow slopes, we generated the contour density plots described by *Maurer et al. (2012)* using the same parameters as the sequence path length analysis. We simulated 100 trials for each running speed. We then divided these 100 trials into 10 subsets of 10 and applied the contour analysis to each subset. We fitted the fast slope to the 95% contour of the central theta peak, and measured the slow slope as the line joining the maximum of the top and bottom peaks of the central 3. We averaged over the results from each subset to obtain the final value. The analytical value for the fast slope in the limit of high phase locking is $FS = v_p/(360f_\theta)$, where the denominator arises due to the normalization to cm/deg in the analysis of *Maurer et al. (2012)*. Similarly for zero phase locking, $FS = v/(360f_\theta)$. The analytical value for the slow slope is independent of phase locking, $SS = v/(360f_\theta)$. Upper and lower bounds for the slow slope were therefore fitted assuming the reported running speed is accurate, and that the LFP theta frequency is in the range 4 Hz $< f_\theta <$ 12 Hz.

## Shuffling analyses

To reproduce the results of *Foster and Wilson (2007)*, we generated data from 1000 theta cycles, each with a running speed drawn from the same distribution as for the prediction analysis. Following the protocol outlined by *Foster and Wilson (2007)*, we found the set of all spike phases for each cell when the rat was at each position and analyzed events defined as 40 ms windows around firing rate peaks. Spike phases were calculated by interpolation between LFP theta peaks. For the shuffling analysis, each spike in an event was replaced by another spike taken from the same cell while the animal was at the same location. The new spike time was then calculated from its phase by interpolation between the closest two LFP theta troughs of the original spike, as reported in the original study. 100 such shuffles were performed for each event, and the correlation between cell rank order and spike times was calculated in each case.

For the corrected shuffling procedure, we followed the methods of *Foster and Wilson (2007)* but with the following adjustments: the correlations between spike times and place field rank order within an event calculated in the original study were replaced with circular-linear correlations between spike phase and place field peaks in order to remove issues arising from conversion between spike time and spike phase (*Kempter et al., 2012*); a minimum running speed of 20 cm/s and a maximum running speed of 100 cm/s were imposed; the LFP phase was measured using a Hilbert transform rather than a linear interpolation between theta peaks; spikes that occured more than 50 cm away from the place field peak were discarded from the analysis. The circular-linear correlation requires a mild restriction of the range of possible regression slopes between the circular and linear variables, which in this case describes the distance traveled by a theta sequence within a theta cycle (*Kempter et al., 2012*). We set this range as 25–80 cm/cycle, that is, around the size of a place field. For simulations using this shuffling procedure, we simulated 300 cells in each session on a linear track and randomly sampled 15 of these for further analysis. We again simulated 10 sessions with 28 laps each, for which the number of detected events was similar to that of the experimental dataset. We generated a large number of such datasets in order to obtain a distribution of shuffling test results to compare against the experimental dataset.

## Dependent and independent cells

To reproduce the results of *Dragoi and Buzsáki (2006)*, we simulated population activity on a linear track. To recreate the experimental conditions of *Dragoi and Buzsáki (2006)*, we set the track length as 250 cm and simulated 8 sessions (i.e., four animals by two running directions), each with 25 place cells. As the original experiment consisted of continuous locomotion around a rectangular track, we wrapped the boundaries of the linear track and simulated continuous sessions rather than single laps. Place fields were randomly distributed over the track following a uniform distribution. Running speed on each lap was drawn from the same distribution as the prediction and shuffling analyses. Phase locking was set to 0.5. We calculated the dependent and independent cell pairs following the methods of *Dragoi and Buzsáki (2006)*, which uses temporal bins of 2 s to calculate firing rate correlations and a shuffling procedure to find significantly correlated cells.

*Dragoi and Buzsáki (2006)* did not state the number of dependent and independent cell pairs obtained from their analysis. Therefore, to compare the results of our simulations to their experimental data, we estimated the number of points in their CCG-lag plot for dependent and independent cell pairs (their Figure 3B) and compared the result to the same measure in our simulations. CCG plots were calculated using the methods described in *Dragoi and Buzsáki (2006)*. Using this method, we found that 33% of cell pairs were dependent compared to an estimated 30–35% in *Dragoi and Buzsáki (2006)*.

To calculate the reliability of temporal lags between dependent and independent pairs, *Dragoi and Buzsáki (2006)* took the central cloud of the CCG-lag vs place field distance scatter plot (their Figure 2A) and calculated the correlation between these two variables. However, the method for isolating the central cloud from the surrounding clusters was not disclosed. Without this information, we were unable to reproduce this analysis.

To test for differences between place field separations of dependent and independent cell pairs, we again considered only cell pairs whose CCG lags passed the inclusion criteria (as described in *Dragoi and Buzsáki (2006)*). We compared the vectors of cell pair separations for each group.

## Decoding individual sequences

To reproduce the results of *Gupta et al. (2012)*, we used the significant sequence testing protocol and Bayesian decoding algorithm described therein, with spatial binning set as 3.5 cm, as in the original study. Briefly, the significant sequence testing analysis tests if population activity within a theta cycle has significant sequential structure, whereas the Bayesian decoding algorithm generates a time by space probability distribution which is used to decode the ahead and behind lengths represented by the theta sequence. For *Figure 6A*, we varied phase locking and running speed independently and generated spiking data for each pair of values. In the simulations used to generate *Figure 6*, we assumed that the number of spikes fired per theta cycle does not vary with running speed, as such a dependence introduces an additional change of the decoded sequence path length with running speed. In order to best match the fraction of theta cycles with three or more cells active reported by *Gupta et al. (2012)*, each simulated theta cycle contained 12 place

cells with place fields randomly distributed over a region of space 94.5 cm ahead or behind the rat. We then applied the significant sequence detection methods for each resulting data set to obtain the fraction of significant sequences in each case. For *Figure 6B*, we used $k = 0.5$ and generated 1000000 theta cycles, each with a running speed drawn from a normal distribution with mean 30 cm/s and standard deviation 10 cm/s. Running speeds less than 10 cm/s were discarded and the remaining theta cycles were tested for significant sequential structure. For *Figure 6C,D*, we applied the Bayesian decoding algorithm to these significant sequences in order to calculate the path length, ahead length, and behind length. In addition, we applied the same analysis to another dataset simulated with $k = 0$.

### Remapping simulations

To simulate remapping in the coordinated assembly model, we simulated spiking activity for a population of 300 cells on a linear track with weights as described in *Supplementary file 1*, Appendix: A4. To simulate the remapped population, we left this set of weights intact but randomly reassigned the place and phase fields of each cell, such that phase coding and rate coding were perfectly remapped but peer interactions were preserved between environments.

To simulate remapping in the linear phase coding model, we assumed that phase lags were preserved between environments. The remapped population was simulated by randomly permuting the place field centers of cells while leaving the phase fields of each cell intact.

To simulate remapping in the sigmoidal phase coding model, we assumed that the field of elevated frequency is locked to the place field before and after remapping. Hence, place fields were randomly permuted and the single cell frequency was defined to increase within the new place field.

### Acknowledgements

This work was supported by the EPSRC, BBSRC, and MRC. We thank Gyuri Buzsaki, Kamran Diba, and Iris Oren for helpful comments on the manuscript. We are grateful for the provision of experimental data, made freely available at crcns.org (*Mizuseki et al., 2014*).

## Additional information

### Funding

| Funder | Author |
| --- | --- |
| Engineering and Physical Sciences Research Council (EPSRC) | Angus Chadwick |
| Biotechnology and Biological Sciences Research Council (BBSRC) | Matthew F Nolan |

The funders had no role in study design, data collection and interpretation, or the decision to submit the work for publication.

### Author contributions

AC, Conception and design, Acquisition of data, Analysis and interpretation of data, Drafting or revising the article; MCWR, MFN, Conception and design, Analysis and interpretation of data, Drafting or revising the article

### Author ORCIDs

Matthew F Nolan, http://orcid.org/0000-0003-1062-6501

## Additional files

### Supplementary file

• Supplementary file 1. Mathematical appendices. This file includes all mathematical methods and derivations pertaining to the models described in the main text.

## Major dataset

The following previously published dataset was used:

| Author(s) | Year | Dataset title | Dataset ID and/or URL | Database, license, and accessibility information |
|---|---|---|---|---|
| Mizuseki K, Diba K, Pastalkova E, Teeters J, Sirota A, Buzsaki G | 2014 | Multiple single unit recordings from different rat hippocampal and entorhinal regions while the animals were performing multiple behavioral tasks | http://dx.doi.org/10.6080/K09G5JRZ | Publicly available at Collaborative Research in Computational Neuroscience (http://crcns.org). |

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
