## [Decision Letter]

[Editors’ note: this article was originally rejected after discussions between
the reviewers, but the authors were invited to resubmit after an appeal against the
decision.]

Thank you for choosing to send your work entitled “Independent Theta Phase Coding
Accounts for CA1 Population Sequences and Enables Flexible Remapping” for
consideration at *eLife*. Your full submission has been evaluated by Eve
Marder (Senior editor) and three peer reviewers, one of whom is a member of our Board of
Reviewing Editors, and the decision was reached after long discussions between the
reviewers. We regret to inform you that your work will not be considered further for
publication.

In summary, although there was some enthusiasm among the reviewers for the work, this
was tempered by concerns regarding insufficient support provided for the strong claims
being made. Specifically, it was felt that this work does not represent a significant
advance in the field without a more substantial analysis to support the authors’
claim that the experimental evidence is consistent with independent generation. Details
of major substantive concerns raised by the reviewers are provided below for your
consideration.

Reviewer #1:

The authors propose an independent coding hypothesis (as compared to a coordinated
assembly hypothesis), in which an essential difference lies in currently active
assemblies not determining future ones in the independent hypothesis case. An addition
to existing phenomenological models (Geisler et al.) is the addition of spike generation
(via an inhomogeneous Poisson process).

1) Given that a 'pacemaker' theta rhythm is included, it does not seem
quite appropriate to talk about the *generation* of an emergent
population theta rhythm. And, how should this be interpreted in light of the mentioned
alternative of “populations*…* generate their own theta
frequency*…*”, as stated in the seventh paragraph of the
subsection “Independent coding accounts for phase sequences”, in the
Results section?

2) It is not clear to me what the new and experimentally testable predictions are? Given
that this is mentioned in the Introduction and Discussion, it should be clearly
delineated. When it appears in the Discussion, subsequent sentences are more about the
differences between the different hypotheses and its consequences, and how the
independent hypothesis is better etc., but not what is 'experimentally
testable' (and feasible?). In earlier parts of the manuscript, there is mention
of optimal peer prediction timescale depending on phase locking, running speed
dependencies etc.

So, what sorts of (feasible) experimental tests are being suggested? And what explicit
predictions (differences from other hypotheses, coordinated assembly) should one look
for? It would help the reader if this was more clearly set down, rather than the
statement of “Important future tests of the independent
coding*…*”, in the fourth paragraph of the Discussion
section.

Since several experimental studies exist, the work as presented does not make it clear
what 'new' experiments need to be done to support and distinguish this
proposed independent hypothesis.

Reviewer #2:

This modeling study explores the population-level implications of the assumption that
CA1 cells phase precess independently.

The main claim that the model “is sufficient to explain the organization of CA1
population activity during theta states” is a bold one, because several
experimental studies argue strongly against independence. The authors re-examine some of
these findings, providing useful new insights in the process. However, I do not think
this main claim is adequately supported. In particular:

1) Some key pieces of evidence against independence are not considered: a) [14] show that cells with
correlated firing rates across laps have more reliable phase lags within theta cycles.
If the authors model correlated firing rates using the Nspikes parameter, it is not
obvious that an increased sequence compression index would result.

b) Schmidt et al. (J Neurosci, 2009) found that on a single pass through a place field,
CA1 cells tend to precess through only about 180 degrees, and that the full 360-degree
range is only obtained after averaging across passes. Thus, a single theta sequence does
not randomly sample the 360 degree range as I believe would be the prediction from the
authors' model. In addition, single trial sequences drawn from the average
place-phase fields were a poor match to the observed sequences, a direct and strong test
of independence.

c) [25] show that forward and
backward sequence length are anticorrelated; this is not the distribution that would
result from the authors' model. In addition, the relationship between velocity
and sequence length reported here in Figure 6 is
the opposite relationship from that shown in the Gupta paper (their Figure 1c).

2) The authors argue that some pertinent pieces of evidence, although originally
advanced as evidence against independence, are in fact consistent with results from
their model. However, in both cases (27; 20) the
authors' argument hinges on a technical point in the original reports. It thus
remains an open, empirical issue whether or not in those papers the main result would
hold if the analyses were performed following the revised method the authors
suggest.

To expand: in the case of Harris et al. it is shown that including more accurate
(direction-dependent) phase fields into the analysis of data generated by the model
improves peer prediction. However, this leaves open the empirical issue of when tested
on actual data, adding peers would further improve the prediction. In the case of Foster
and Wilson, the authors find that shuffling their model data as reported in that paper
reduces correlations indicative of theta sequences, demonstrating that this result can
in fact be obtained from independent neurons. Interestingly, the authors note that the
method reported in the original paper may be modified to be a stronger test of
non-independence. As reported it remains to be seen whether if, with this modified
shuffling procedure, correlations would be reduced.

Given the above issues, I think the authors should align their interpretation with what
they have shown (and not shown!). Overall I am enthusiastic about several aspects of the
work: the model is compact and intuitive, providing not only satisfying insight, but
also novel applications to experimental data sets with arbitrary speed profiles. It is
thus a useful tool that sharpens the interpretation of such data sets, and suggests new
analyses and experiments moving forward. The exploration of the relationship between
theta sequences and remapping is thoughtful and generates useful testable
predictions.

Reviewer #3:

This manuscript describes a model of hippocampal cell activity in which phase precession
arises due to “traveling waves” within CA1. The authors first define their
model, and then demonstrate that this model matches a number of data sets from
experiments examining phase precession.

I think this model is clearly described and easy to understand. It should be easily
generalized to other areas that demonstrate phase precession (for example CA3). However,
the model is mainly descriptive, meaning it describes and characterizes the phenomenon
of phase precession, but does not provide any new insights into the underlying
mechanisms of phase precession, nor any new insights into how information is encoded in
the hippocampus. The equations used to predict the firing rates of CA1 cells are very
similar to equations used in previous studies of phase precession (the authors even cite
some of these references). While I do think that having a clear description of phase
precession and the implications that the existence of this phenomenon has on activity
patterns within CA1 would be useful to the field, such a discussion is almost more
appropriate for a review paper rather than a research article. Also, the manuscript is
rather dense with concepts that are specific to temporal coding in the hippocampus. I am
not sure how understandable this paper would be to those outside of this field.

Major comments:

1) Through their comparisons of the model with experimental data, the authors have
provided an excellent review of the literature concerning hippocampal phase precession.
I suspect, however, that sections of this paper may be incomprehensible to those not
specifically working on phase precession. For example, I would have rather seen a more
in-depth review of peer-predictions and its implications, rather than what felt like a
cursory explanation and a citation of the Harris paper.

2) There is not much discussion of previous models of phase precession, although many of
the experimental results used by the authors to support their model is
predicted/explained by other models as well. For example, [22] already demonstrated that the LFP theta
rhythm can from a population of neurons oscillating faster than the theta-frequency.
Although this paper is cited, I feel more attention should be paid to the analytical
results of that paper rather than just focusing on the experimental data.

3) Along the same lines, it would have been nice to see at least some discussion of
previous models of phase precession. While an exhaustive comparison is perhaps beyond
the scope of this manuscript, O'Keefe's dual oscillator model is a classic
and should at least be discussed.

---

## [Author Response]

[Editors’ note: this article was originally rejected after discussions between
the reviewers, but the authors were invited to resubmit after an appeal against the
decision.]

*In summary, although there was some enthusiasm among the reviewers for the work,
this was tempered by concerns regarding insufficient support provided for the strong
claims being made. Specifically, it was felt that this work does not represent a
significant advance in the field without a more substantial analysis to support the
authors’ claim that the experimental evidence is consistent with independent
generation. Details of major substantive concerns raised by the reviewers are
provided below for your consideration*.

We thank the reviewers and editor for their helpful comments. In recognizing the initial
concerns identified by the reviewers we have carried out substantial additional
analyses: a) we now compare predictions from independent coding with a coordinated
coding model, and b) we now compare analysis of experimental data with model
predictions. Our new analysis provides further support to our initial conclusions. We
outline these and further changes in the response below.

Reviewer #1:

*The authors propose an independent coding hypothesis (as compared to a
coordinated assembly hypothesis), in which an essential difference lies in currently
active assemblies not determining future ones in the independent hypothesis case. An
addition to existing phenomenological models (Geisler et al.) is the addition of
spike generation (via an inhomogeneous Poisson process)*.

We note that while our model, like that of Geisler et al., addresses phase precessing
assemblies at a phenomenological level, it differs conceptually in a number of important
ways. First, our model allows an analysis of the spatiotemporal patterns of population
activity, whereas Geisler et al. only investigated the temporal dynamics of single unit
and population average activity. This is important because it allows analysis of theta
sequences at the population level, which is central to the new advances made by our
study. Second, our model generates realistic activity at arbitrary running speeds, while
the fixed phase lags assumed by Geisler et al. are inconsistent with experimental data
if running speeds are allowed to vary. Third, our model allows systematic variation of
the phase locking of cells against the theta rhythm, leading to novel predictions for
sequence properties, including a dependence of the decoded sequence path length and
propagation speed on phase locking (Figure 5 and
Figure 5—figure supplement 1 in our
original and revised manuscripts). We appreciate we may not have made these important
conceptual differences clear in our initial manuscript and have addressed this in the
revised submission (across the subsection headed “Independent phase coding
generates traveling waves”, in the Results section).

*1) Given that a 'pacemaker' theta rhythm is included, it does not
seem quite appropriate to talk about the* generation *of an emergent
population theta rhythm. And, how should this be interpreted in light of the
mentioned alternative of “populations… generate their own theta
frequency…”, as stated in the seventh paragraph of the subsection
“Independent coding accounts for phase sequences”, in the Results
section?*

We appreciate the reviewer's point and believe it perhaps reflects a lack of
clarity on our part in the original manuscript. Thus, while the origin of the theta
frequency oscillation is not central to our main conclusions, the reviewer identifies an
element of our model that, because it is conceptually similar to that of Geisler et al.,
we perhaps did not explain sufficiently clearly. In our single cell model, we define
neurons to precess in phase against a reference theta rhythm. As a result, each neuron
oscillates at a velocity-dependent frequency which is always higher than that of the
reference theta. Regardless of velocity, however, we find that the global population
activity oscillates at the same frequency as the reference theta, i.e. at a lower
frequency than each individual cell. Our use of the word “generate” is
restricted to this scenario, where the network theta is “generated” from
the sum of the higher frequency oscillations in each neuron. In the revised manuscript
we have clarified this point (second paragraph of the subsection “Independent
phase coding generates traveling waves”, in the Results).

*2) It is not clear to me what the new and experimentally testable predictions
are? Given that this is mentioned in the Introduction and Discussion, it should be
clearly delineated. When it appears in the Discussion, subsequent sentences are more
about the differences between the different hypotheses and its consequences, and how
the independent hypothesis is better etc., but not what is 'experimentally
testable' (and feasible?). In earlier parts of the manuscript, there is
mention of optimal peer prediction timescale depending on phase locking, running
speed dependencies etc*.

*So, what sorts of (feasible) experimental tests are being suggested? And what
explicit predictions (differences from other hypotheses, coordinated assembly) should
one look for? It would help the reader if this was more clearly set down, rather than
the statement of “Important future tests of the independent coding…
”, in the fourth paragraph of the Discussion section*.

*Since several experimental studies exist, the work as presented does not make it
clear what 'new' experiments need to be done to support and distinguish
this proposed independent hypothesis*.

We appreciate this was a major weakness of the previous manuscript and have carried out
substantial new simulations and analysis of experimental data to address the point at
length. We previously identified predictions that distinguish different scenarios for
independent coding, but we did not make explicit predictions for analyses that would
distinguish independent from coordinated coding. We also did not compare predictions
from independent and coordinated coding models directly with experimental data and, as
the reviewer notes, we did not distinguish predictions that require new experiments. We
have addressed these issues as follows:

a) To address the question of how the independent coding hypothesis might be empirically
distinguished from the coordinated coding hypothesis, we have developed an additional
model and performed extensive additional simulations and analyses. The additional model
includes interactions between cells within the population in order to simulate data
under the coordinated assembly hypothesis (Figure 3—figure supplement 1 in the revised manuscript), while the additional
simulations compare the behavior of the independent coding and coordinated assembly
models when subjected to statistical tests of independence. In particular, we compared
the performance of a shuffling analysis (adapted from [20]; see Figure 5—figure supplement 2 in the revised manuscript) and a prediction
analysis (adapted from [27];
please see Table 1 and Figure 4—figure supplement 1 and Figure 4—figure supplement 2 in the revised
manuscript). We find in both cases that spike patterns generated by independent coding
and coordinated assembly models can be distinguished by the shuffling analysis and by
the prediction analysis. We are able to estimate the statistical power of each analysis
method given assumptions about the effect size and the number of recorded neurons.

We include these new results in the Results section of the revised manuscript (please
see the subsections: “Assembly coordination stabilizes sequential activation
patterns”, “Independent coding accounts for apparent peer-dependence of
CA1 activity”, and “Independent coding accounts for phase
sequences”).

We also clarify novel predictions requiring new data, including predictions involving
membrane potential oscillations and place field remapping (subsection “Linear
phase coding constrains global remapping” in the Results and paragraph three in
the Discussion). We note here that, in addition to our previous predictions, our new
coordinated assembly model has allowed the additional prediction that phase precession
would be severely disrupted following remapping if CA1 assemblies were generated by
coordinated coding (Figure 7—figure supplement 1 and subsection “Linear phase coding constrains global
remapping”, in the Results of the revised manuscript).

b) Having demonstrated that these new analyses have the statistical power to distinguish
independent from coordinated data, we applied these analyses to experimental data (for
details of these data, see [52]). For both tests, the experimental data favor the independent coding
hypothesis (please see the subsections “Independent coding accounts for apparent
peer-dependence of CA1 activity” and “Independent coding accounts for
phase sequences” of the Results, Table 1, Figure 4—figure supplement 1 and Figure 4—figure supplement 2 and Figure 5—figure supplement 2 in the revised manuscript). We believe this new analysis provides very
substantial new evidence which supports our original conclusions.

Reviewer #2:

*This modeling study explores the population-level implications of the assumption
that CA1 cells phase precess independently*.

*The main claim that the model “is sufficient to explain the organization
of CA1 population activity during theta states” is a bold one, because several
experimental studies argue strongly against independence. The authors re-examine some
of these findings, providing useful new insights in the process. However, I do not
think this main claim is adequately supported. In particular*:

*1) Some key pieces of evidence against independence are not considered:
a)*
[14]
*show that cells with correlated firing rates across laps have more reliable
phase lags within theta cycles. If the authors model correlated firing rates using
the Nspikes parameter, it is not obvious that an increased sequence compression index
would result*.

We appreciate the suggestion, but unfortunately we find that certain steps of the
analysis reported by Dragoi et al. were not exactly reproducible due to a lack of
information in their study. For example, the method for isolating the central cloud
(their Figure 2B) from the overall CCG plot (their Figure 2A) was not disclosed.
Nevertheless, in attempting to reproduce their methods as closely as possible, we found
that their key results could be accounted for by independent coding. First, when
analyzing the correlations in lap by lap firingrates using the methods of Dragoi et al.,
we found a similar number of apparently dependent cell pairs as the original study,
despite the absence of true firingrate correlations within the simulated data. Hence,
the analysis used by Dragoi et al. artificially separates homogeneous populations of
place cells into apparently dependent and independent cell pairs. Second, these
dependent and independent cell groups displayed different spatial distributions of
firing rate fields. This suggests that the effects reported by Dragoi et al. might
result from a sampling bias introduced by the separation of a homogeneous population of
cells into dependent and independent groups. Thus, as far as we can tell, the results
reported by Dragoi et al. are consistent with the independent coding hypothesis. We
report these new analyses in the revised manuscript (paragraph seven of subsection
“Independent coding accounts for phase sequences”, in the Results). We
hope the reviewers and editors will also recognize the difficulty in making comparisons
to previous work where that work has not been documented to a level where it can be
reproduced.

*b) Schmidt et al. (J Neurosci, 2009) found that on a single pass through a place
field, CA1 cells tend to precess through only about 180 degrees, and that the full
360-degree range is only obtained after averaging across passes. Thus, a single theta
sequence does not randomly sample the 360 degree range as I believe would be the
prediction from the authors' model. In addition, single trial sequences drawn
from the average place-phase fields were a poor match to the observed sequences, a
direct and strong test of independence*.

We note that the data of Schmidt et al. do not provide evidence for or against
independent coding. This is because Schmidt et al. did not analyze sequences, but only
single unit phase precession. Hence, while their results suggest a more complex single
cell phase code than that included in our model, they cannot reveal coordination between
cells as this would require an analysis of ensemble activity. Our single cell model can
be readily extended to incorporate more complex single cell phase codes while
maintaining independence between cells in the population. We now discuss this issue in
the manuscript and include an additional appendix detailing a model which includes
trial-by-trial single cell coding properties resembling those described by Schmidt et
al. while maintaining independence between cells (final paragraph of subsection headed
“Single Cell Coding Model”, in the Results section, and Appendix: A2).

*c)*
[25]
*show that forward and backward sequence length are anticorrelated; this is not
the distribution that would result from the authors' model. In addition, the
relationship between velocity and sequence length reported here in*
Figure 6
*is the opposite relationship from that shown in the Gupta paper (their*
*Figure 1c**)*.

While we agree with the reviewer that our previous presentation of the independent
coding model did suggest a difference to the observations in Gupta et al., it does not
follow that the Gupta et al data are inconsistent with independent coding. We have
performed additional simulations using the Gupta protocol which show that, if the number
of cells simulated is sufficiently small as to match the number of cells per theta cycle
reported by Gupta et al., the anticorrelation between ahead and behind length arises
naturally due to the sequence selection criteria. Importantly, these results are fully
reproducible for realistic values of phase locking and also for zero phase locking,
where no theta sequences exist in the data (revised manuscript Figure 6). We further note that the relationship between
velocity and sequence path length that we presented in our original manuscript was a
consequence of our assumed change in firingrate as a function of running speed. Further
simulations in which the number of spikes per theta cycle does not vary with running
speed produce a relationship similar to that reported by Gupta et al. (revised
manuscript Figure 6). Thus, the Gupta et al.
data are fully consistent with the independent coding model. We make these issues clear
in the revised manuscript (paragraph eight of the subsection “Independent coding
accounts for phase sequences”, in the Results section).

*2) The authors argue that some pertinent pieces of evidence, although originally
advanced as evidence against independence, are in fact consistent with results from
their model. However, in both cases (*[27]*;*
[20]*) the authors' argument hinges on a technical
point in the original reports. It thus remains an open, empirical issue whether or
not in those papers the main result would hold if the analyses were performed
following the revised method the authors suggest*.

*To expand: in the case of Harris et al. it is shown that including more accurate
(direction-dependent) phase fields into the analysis of data generated by the model
improves peer prediction. However, this leaves open the empirical issue of when
tested on actual data, adding peers would further improve the prediction. In the case
of Foster and Wilson, the authors find that shuffling their model data as reported in
that paper reduces correlations indicative of theta sequences, demonstrating that
this result can in fact be obtained from independent neurons. Interestingly, the
authors note that the method reported in the original paper may be modified to be a
stronger test of non-independence. As reported it remains to be seen whether if, with
this modified shuffling procedure, correlations would be reduced*.

*Given the above issues, I think the authors should align their interpretation
with what they have shown (and not shown!). Overall I am enthusiastic about several
aspects of the work: the model is compact and intuitive, providing not only
satisfying insight, but also novel applications to experimental data sets with
arbitrary speed profiles. It is thus a useful tool that sharpens the interpretation
of such data sets, and suggests new analyses and experiments moving forward. The
exploration of the relationship between theta sequences and remapping is thoughtful
and generates useful testable predictions*.

We appreciate these points reflected substantial weaknesses in the previous manuscript.
As detailed in our response to Reviewer 1, we have now performed extensive additional
simulations and analyses which address these issues directly and in full. In particular,
we show through simulations that our improved tests can successfully distinguish between
coordinated and independent coding (please see the subsections “Independent
coding accounts for apparent peer-dependence of CA1 activity” and
“Independent coding accounts for phase sequences”, in the Results section
of the revised manuscript), and we show that the results of these tests when applied to
experimental data suggest independent coding rather than coordinated assemblies (see
revised manuscript Table 1, Figure 4—figure supplement 1 and Figure 4—figure supplement 2 and
Figure 5—figure supplement 2).
Our new simulations and analysis therefore provide further and we believe very
substantial support to the independent coding hypothesis.

Reviewer #3:

*This manuscript describes a model of hippocampal cell activity in which phase
precession arises due to “traveling waves” within CA1. The authors
first define their model, and then demonstrate that this model matches a number of
data sets from experiments examining phase precession*.

*I think this model is clearly described and easy to understand. It should be
easily generalized to other areas that demonstrate phase precession (for example
CA3). However, the model is mainly descriptive, meaning it describes and
characterizes the phenomenon of phase precession, but does not provide any new
insights into the underlying mechanisms of phase precession, nor any new insights
into how information is encoded in the hippocampus. The equations used to predict the
firing rates of CA1 cells are very similar to equations used in previous studies of
phase precession (the authors even cite some of these references). While I do think
that having a clear description of phase precession and the implications that the
existence of this phenomenon has on activity patterns within CA1 would be useful to
the field, such a discussion is almost more appropriate for a review paper rather
than a research article. Also, the manuscript is rather dense with concepts that are
specific to temporal coding in the hippocampus. I am not sure how understandable this
paper would be to those outside of this field*.

We disagree with the Reviewer 3's suggestion that the model is descriptive and
does not provide new insights. The reviewer's comments focus on our
phenomenological model of phase precession in single cells. While this is in fact novel,
as we make clear in our response to Reviewer 1 above, the conceptual importance of our
work comes from our investigation of the population level activity predicted by this
model. In this respect, it is only necessary that our model provides a good account of
phase precession in single cells. We do not make any claims about mechanisms of phase
precession. Given this misconception, we highlight again the key conceptual advances
made by our study.

While considerable previous work has argued that population activity in CA1 during theta
states involves coordinated coding, our model demonstrates that experimental evidence to
support this conclusion can be fully accounted for by independent coding. We then use
the model to develop novel insights into the implications of independent coding for
place cell remapping. Thus, our manuscript provides a fundamentally different conception
of population activity in CA1 to previous studies. Because theta activity in CA1 is
coordinated with other circuits including prefrontal cortex and entorhinal cortex, our
results have wide reaching implications for neural coding in general.

Our new simulations identify experimentally testable predictions that distinguish
population activity under coordinated and independent coding scenarios. By comparison of
these predictions to experimental data we now provide strong evidence in support of
independent coding. We provide novel predictions for the consequences of different
independent coding models for remapping of place representations. We have now extended
this analysis to show that coordinated and independent coding fundamentally differ in
their capabilities and limitations. Independent coding offers a massively increased
ability to encode multiple environments, while coordinated coding provides a mechanism
by which robust sequential activity can be generated despite the noisy intrinsic
properties of individual place cells.

Thus, the models and analysis introduced by our study offer fundamental insights into
both information processing and coordination of spike timing in hippocampal populations.
In our revised manuscript we take care to make these novel insights much clearer to the
reader.

*Major comments*:

*1) Through their comparisons of the model with experimental data, the authors
have provided an excellent review of the literature concerning hippocampal phase
precession. I suspect, however, that sections of this paper may be incomprehensible
to those not specifically working on phase precession. For example, I would have
rather seen a more in-depth review of peer-predictions and its implications, rather
than what felt like a cursory explanation and a citation of the Harris
paper*.

We have performed extensive additional peer prediction simulations on both simulated and
experimental data (see comments above). Accordingly, this section of the manuscript has
been expanded and a more in-depth explanation is included (subsection
“Independent coding accounts for apparent peer-dependence of CA1
activity”, in the Results section).

*2) There is not much discussion of previous models of phase precession, although
many of the experimental results used by the authors to support their model is
predicted/explained by other models as well. For example,*
[22]
*already demonstrated that the LFP theta rhythm can from a population of neurons
oscillating faster than the theta-frequency. Although this paper is cited, I feel
more attention should be paid to the analytical results of that paper rather than
just focusing on the experimental data*.

While we agree that several previous models of phase precession can account for the same
phenomenological results as our model at the single-cell level, the purpose of our study
is not to explain the mechanisms of single-cell phase precession but rather to
understand the emergence of population activity. We now carefully explain the
similarities and differences from the Geisler model, which we detailed above in our
response to Reviewer 1.

*3) Along the same lines, it would have been nice to see at least some discussion
of previous models of phase precession. While an exhaustive comparison is perhaps
beyond the scope of this manuscript, O'Keefe's dual oscillator model is
a classic and should at least be discussed*.

In the Discussion we outline which previous models of phase precession could provide a
mechanistic basis for our single cell coding model and which models would instead imply
coordinated assemblies. We now pay specifically mention the dual oscillator and other
models in the Discussion section of the updated manuscript. Since the cellular
mechanisms of phase precession are not a focus of our study we do not discuss these at
length.